# DriveVLA-W0: World Models Amplify Data Scaling Law in Autonomous Driving

**Yingyan Li**[1*]  **Shuyao Shang**[1*]  **Weisong Liu**[1*]  **Bing Zhan**[1*]  **Haochen Wang**[1*]
**Yuqi Wang**[1]  **Yuntao Chen**[1]  **Xiaoman Wang**[2]  **Yasong An**[2]
**Chufeng Tang**[2]  **Lu Hou**[2]  **Lue Fan**[1✉]  **Zhaoxiang Zhang**[1✉]

[1]NLPR, Institute of Automation, Chinese Academy of Sciences (CASIA)
[2]Yinwang Intelligent Technology Co. Ltd.

{liyingyan2021,shangshuyao2024,liuweisong2024,zhanbing2024}@ia.ac.cn
{lue.fan, zhaoxiang.zhang}@ia.ac.cn

Code: https://github.com/BraveGroup/DriveVLA-W0

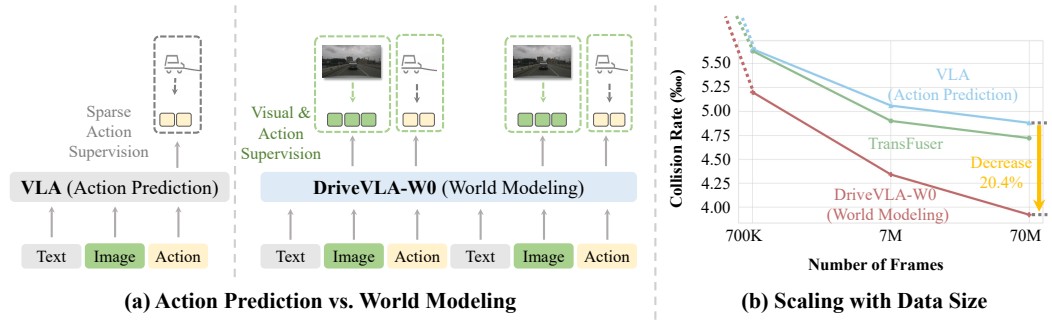

**Figure 1: World modeling as a catalyst for VLA data scalability.** (a): Unlike standard VLAs trained solely on action supervision, our DriveVLA-W0 is trained to predict both future actions and visual scenes. (b): This world modeling task provides a dense source of supervision, enabling our model to better harness the benefits of large-scale data.

## Abstract

Scaling Vision-Language-Action (VLA) models on large-scale data offers a promising path to achieving a more generalized driving intelligence. However, VLA models are limited by a "supervision deficit": the vast model capacity is supervised by sparse, low-dimensional actions, leaving much of their representational power underutilized. To remedy this, we propose **DriveVLA-W0**, a training paradigm that employs world modeling to predict future images. This task generates a dense, self-supervised signal that compels the model to learn the underlying dynamics of the driving environment. We showcase the paradigm's versatility by instantiating it for two dominant VLA archetypes: an autoregressive world model for VLAs that use discrete visual tokens, and a diffusion world model for those operating on continuous visual features. Building on the rich representations learned from world modeling, we introduce a lightweight action expert to address the inference latency for real-time deployment. Extensive experiments on the NAVSIM v1/v2 benchmark and a 680x larger in-house dataset demonstrate that DriveVLA-W0 significantly outperforms BEV and VLA baselines. Crucially, it amplifies the data scaling law, showing that performance gains accelerate as the training dataset size increases.

## 1 Introduction

The promise of scaling laws (Kaplan et al., 2020; Zhai et al., 2022; Baniodeh et al., 2025; Dehghani et al., 2023) presents an attractive path toward more generalized driving intelligence, with the hope that petabytes of driving data can be harnessed to train powerful foundation models. In the current

landscape, two dominant paradigms exist. On one side are specialized models (Hu et al., 2022; Jiang et al., 2023) centered around Bird's-Eye-View (BEV) representations (Li et al., 2022; Huang et al., 2021). These models are built upon carefully designed geometric priors, which, while effective for driving-specific tasks, make it less straightforward to leverage non-driving datasets. In addition, their relatively compact architectures may constrain their potential for large-scale data scalability. In response, Vision-Language-Action (VLA) models (Fu et al., 2025a; Li et al., 2025b; Zhou et al., 2025d) have emerged as a promising alternative. By leveraging large-scale Vision-Language Models (VLMs) (Wang et al., 2024b; Bai et al., 2025) pretrained on internet-scale data, VLAs possess a significantly larger model size and a greater intrinsic potential for scaling.

However, this scaling potential is largely unrealized due to a critical challenge: the immense model size of VLA models is met with extremely sparse supervisory signals. The standard paradigm involves fine-tuning these VLM models solely on expert actions. This tasks the model with mapping high-dimensional sensory inputs to a few low-dimensional control signals (e.g., waypoints). This creates a severe "*supervision deficit*". This deficit prevents the model from learning rich world representations, a fundamental limitation that cannot be overcome by simply increasing the volume of action-only training data. In fact, we observe that without sufficient supervision, large VLA models can even underperform smaller, specialized BEV models.

To address this supervision deficit, we harness the power of world modeling (Li et al., 2024a; Wang et al., 2025; Cen et al., 2025; Chen et al., 2024), integrating it as a dense self-supervised objective to supplement the sparse action signal. By tasking the model with predicting future images, we generate a dense and rich supervisory signal at every timestep. This objective forces the model to learn the underlying dynamics of the environment and build a rich, predictive world representation. To validate the effectiveness of our approach, we implement it across the two dominant VLA architectural families, which are primarily differentiated by their visual representation: discrete tokens versus continuous features. For VLAs that represent images as discrete visual tokens, world modeling is a natural extension. We propose an autoregressive world model to predict the sequence of discrete visual tokens of future images. For VLAs that operate on continuous features, this task is more challenging as they lack a visual vocabulary, making a direct next-token prediction approach infeasible. To bridge this gap, we introduce a diffusion world model that generates future image pixels conditioned on the vision and action features produced at the current frame.

We validate our world modeling approach across multiple data scales, from academic benchmarks to a massive in-house dataset. First, experiments scaling at academic benchmarks reveal that world modeling is crucial for *generalization*, as it learns robust visual patterns rather than overfitting to dataset-specific action patterns. To study true scaling laws, we then leverage a massive 70M-frame in-house dataset, as Figure 1 shows. This confirms our central hypothesis: world modeling amplifies the data scaling law. This advantage stems from the dense visual supervision provided by future frame prediction, creating a qualitative gap that cannot be closed by purely scaling the quantity of action-only data. Finally, to enable real-time deployment, we introduce a lightweight, MoE-based Action Expert. This expert decouples action generation from the large VLA backbone, reducing inference latency to just 63.1% of the baseline VLA and creating an efficient testbed to study different action decoders at a massive scale. This reveals a compelling reversal of performance trends from smaller to larger data scales. While complex flow-matching decoders often hold an advantage on small datasets, we find that this relationship inverts at a massive scale, where the simpler autoregressive decoder emerges as the top performer. Our work makes three primary contributions:

- We identify the "supervision deficit" as a critical bottleneck for scaling VLAs and propose **DriveVLA-W0**, a paradigm that uses world modeling to provide a dense, self-supervised learning signal from visual prediction.

- Our experiments reveal two key scaling advantages of world modeling. First, it enhances generalization across domains with differing action distributions by learning transferable visual representations. Second, on a massive 70M-frame dataset, it amplifies the data scaling law, providing a benefit that simply scaling up action-only supervision cannot achieve.

- We introduce a lightweight MoE-based Action Expert that reduces inference latency to 63.1% of the baseline. Using this expert as a testbed, we uncover a compelling scaling law reversal for action decoders: simpler autoregressive models surpass more complex flow-matching ones at a massive scale, inverting the performance trend seen on smaller datasets.

## 2 RELATED WORK

**VLAs in Autonomous Driving.** The development of VLMs in autonomous driving has progressed through three main stages, evolving from focusing on interpretation to integrated end-to-end VLA architectures. The first stage, language-based scene interpretation models, focused on enhancing interpretability. Models like DriveGPT4 (Xu et al., 2024) used Large Language Models (LLMs) to generate scene explanations and high-level maneuver suggestions, but without producing directly executable actions (Chen et al., 2025; Zhou et al., 2025c). The second stage, modular language-to-action frameworks, aimed to connect high-level commands with low-level control. These approaches (Zhou et al., 2025b; Yuan et al., 2024; Zhang et al., 2025; Arai et al., 2025; Gao et al., 2025) utilize multi-stage pipelines where modules are connected via non-differentiable interfaces, such as discrete textual commands, which prevent gradient back-propagation. The current frontier is End-to-End VLAs (Renz et al., 2025; 2024; Jiang et al., 2025; Hwang et al., 2024; Yang et al., 2025; Zeng et al., 2025), which employ a unified architecture to directly map sensor inputs to trajectories. Within this paradigm, Orion (Fu et al., 2025a) enables holistic reasoning through vision-language instructed action generation. Additionally, models like MindDrive (Fu et al., 2025b) and ReCogDrive (Li et al., 2025b) leverage reinforcement learning to effectively bridge the language-action gap. AutoVLA (Zhou et al., 2025d) tokenizes trajectories into action primitives, enabling a single autoregressive model to learn adaptive reasoning and planning. We follow this paradigm and propose an end-to-end VLA.

**World Models in Driving and Robotics** Research on world models typically follows two distinct philosophies: utilizing them as data synthesizers for simulation, or employing them as an auxiliary objective for representation learning. The first stream focuses on **generation**, aiming to synthesize high-quality driving data. Prominent works like GAIA-1 (Hu et al., 2023), DrivingGPT (Chen et al., 2024), and Doe-1 (Zheng et al., 2024a) generate multi-modal tokens for realistic scenarios, while unified models like Hermes (Zhou et al., 2025a) and UniFuture (Liang et al., 2025) simultaneously perform future generation and 3D scene perception. Conversely, the second stream leverages world modeling as a **self-supervised objective** to enhance representation learning. While VaVAM (Bartoccioni et al., 2025) demonstrates the value of generative video pretraining, it lacks action conditioning. Addressing this, UniVLA (Wang et al., 2025) and WorldVLA (Cen et al., 2025) incorporate actions into autoregressive prediction to learn robust policies. In autonomous driving, LAW (Li et al., 2024a) pioneers this paradigm using a latent world model. Distinct from LAW's latent prediction, our work supervises the model to predict future images, providing a richer and more direct dense learning signal for the VLA backbone.

## 3 METHODOLOGY

Our methodology unfolds in three key steps. First, we establish a **VLA Baseline** to demonstrate the challenges of sparse action-only supervision. Second, we enhance this baseline with **World Modeling**, our primary contribution, which provides dense self-supervision. Building on this, we tackle the inference bottleneck by introducing a lightweight, MoE-based **Action Expert**, ensuring our powerful model achieves real-time performance.

### 3.1 VLA BASELINE

Our Vision-Language-Action (VLA) baseline processes sequences of language instructions ($L_t$), front-view images ($V_t$), and past actions ($A_{t-1}$). To ensure broad applicability, we build variants on the two dominant VLM paradigms: **VLA (VQ)**, which quantizes images into discrete visual tokens for an Emu3-style backbone, and **VLA (ViT)**, which extracts continuous features for a Qwen2.5-VL-style backbone.

**Input Tokenization.** High-level driving language instructions ($L_t$) are processed using the VLM's native tokenizer. For past actions, we use the FAST tokenizer (Pertsch et al., 2025) to convert continuous waypoint trajectories into a sequence of discrete tokens, denoted $A_{t-1}$.

**VLM Backbone.** At each timestep $t$, we form a deeply interleaved input sequence $S_t$ by concatenating multimodal chunks over a history of $H$ steps following Wang et al. (2025); Fan et al. (2025): $S_t = [L_{t-H}, V_{t-H}, A_{t-H-1}, \ldots, L_t, V_t, A_{t-1}]$. This sequence is processed autoregressively by our VLM backbone, for which we select two representative models: Emu3 (8B) (Wang et al., 2024b)

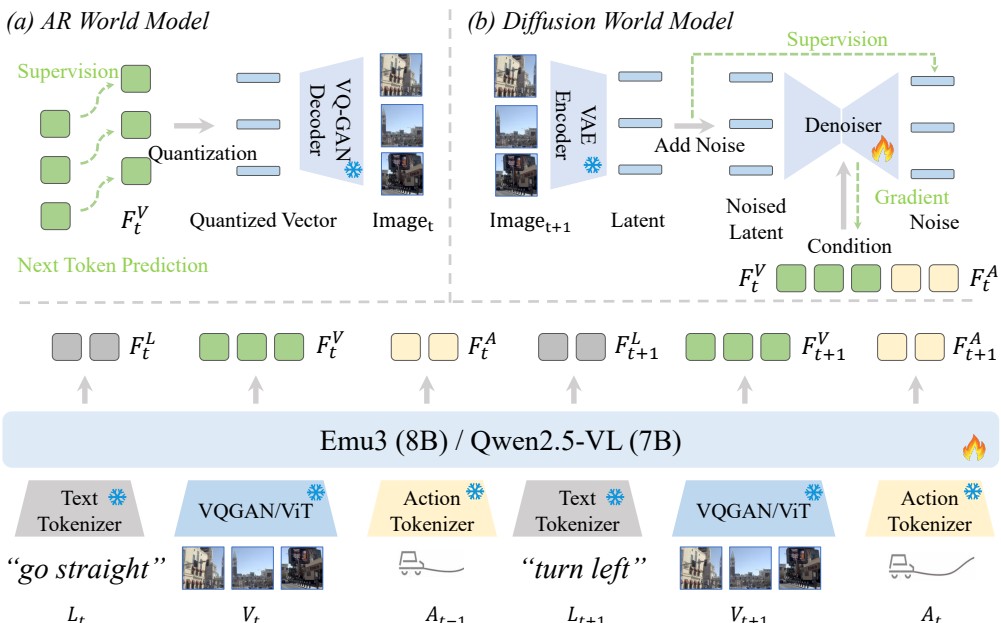

Figure 2: **The architecture of *DriveVLA-W0*,** which achieves world modeling in two ways: (a) an AR World Model that predicts discrete visual tokens, and (b) a Diffusion World Model that denoises latent representations conditioned on multimodal inputs.

to handle discrete visual representations and Qwen2.5-VL (7B) (Bai et al., 2025) for continuous features, using a causal attention mask. The VLM backbone outputs the final-layer hidden states, which are then split according to their respective modalities into language ($F_t^L$), vision ($F_t^V$), and action ($F_t^A$) features.

**Action Prediction.** For training, we optimize the model to predict the ground-truth action token sequence $A_t = (a_1, \ldots, a_M)$ using a standard cross-entropy loss:

$$\mathcal{L}_{\text{Action}} = -\sum_{i=1}^{L} \log P(a_i | S_t, a_{<i}). \tag{1}$$

During inference, the trained model then autoregressively generates a sequence of action tokens conditioned on the context $S_t$. These tokens are subsequently converted back into a continuous waypoint trajectory by the FAST detokenizer (Pertsch et al., 2025).

## 3.2 WORLD MODELING

Prior VLA pipelines typically *only* supervise the model's actions. This yields a sparse supervisory signal, compressing high-dimensional sensory inputs into a few low-dimensional control signals and results in a "supervision deficit". To address this, we introduce world modeling as a powerful self-supervised objective. We implement the world model differently for our two VLA paradigms. For VLAs equipped with a discrete visual vocabulary, we formulate the world model as a next-token prediction task, creating our **AR World Model**. Conversely, for VLAs that operate on continuous visual features, we introduce a **Diffusion World Model** to generate future images in a continuous latent space.

**AR World Model.** Our AR World Model predicts the current visual scene by autoregressively generating its sequence of discrete visual tokens, conditioned on past observations and actions (Figure 2 (a)).

***Training.*** The model learns to autoregressively generate the sequence of visual tokens for the current image, $V_t = (v_1, \ldots, v_N)$, conditioned on the preceding context $S_{<V_t}$. The process is optimized by

minimizing the next-token prediction loss

$$\mathcal{L}_{\text{WM-AR}} = -\sum_{i=1}^{N} \log P(v_i | S_{<V_t}, v_{<i}).$$ (2)

We refer to this complete framework as **DriveVLA-W0 (VQ)**. It is trained jointly by optimizing a weighted sum of the action and AR world model losses: $\mathcal{L}_{\text{Total}} = \mathcal{L}_{\text{Action}} + \alpha \mathcal{L}_{\text{WM-AR}}$, where $\alpha$ is a balancing coefficient.

*Inference.* While the explicit generation of visual tokens is typically bypassed during inference to ensure low latency, this capability remains valuable for visualization purposes. To generate an image, the model autoregressively samples a sequence of visual tokens, which are then passed to the MoVQGAN (Zheng et al., 2022) decoder to render the final image $\hat{I}_t$.

**Diffusion World Model.** Unlike the VQ-based counterpart, our *VLA (ViT)* model lacks a discrete visual vocabulary suitable for next-token prediction. We therefore introduce a Diffusion World Model, which instead provides dense supervision by training a latent diffusion model (Rombach et al., 2021; Wang et al., 2024a) to generate future images conditioned on the VLA's rich output features ($F_t^V, F_t^A$) as Figure 2 shows. This choice to predict the future frame ($\mathbf{I}_{t+1}$) is critical: since the model is conditioned on all present features simultaneously, predicting the future is necessary to learn predictive dynamics rather than simply performing a reconstruction task.

*Training.* This framework learns to predict the future visual scene ($\mathbf{I}_{t+1}$) conditioned on the VLA's current visual and action features ($F_t^V$ and $F_t^A$). Following the standard latent diffusion setup, the model is trained to denoise a noised version of the future image's latent representation. This is optimized via an MSE objective

$$\mathcal{L}_{\text{WM-Diff}} = \mathbb{E}_{z_{t+1}, \epsilon, k} \left[ \| \epsilon - \hat{\epsilon}(z_{t+1,k}, k, F_t^V, F_t^A) \|^2 \right].$$ (3)

where $z_{t+1}$ is the latent of the future image $\mathbf{I}_{t+1}$, $\epsilon \sim \mathcal{N}(0, \mathbf{I})$ is sampled Gaussian noise, $k$ is a random diffusion timestep, and $\hat{\epsilon}$ is the denoiser network trained to predict the noise from the noised latent $z_{t+1,k}$. We term this overall framework **DriveVLA-W0 (ViT)**. It is trained end-to-end by optimizing a joint objective that combines the action prediction loss and the diffusion world model loss: $\mathcal{L}_{\text{Total}} = \mathcal{L}_{\text{Action}} + \beta \mathcal{L}_{\text{WM-Diff}}$, where $\beta$ is a balancing coefficient.

*Inference.* As with the AR model, the diffusion process is bypassed during driving inference to ensure real-time performance. For qualitative analysis, future frames can be generated by running the reverse diffusion process, starting from random noise and conditioning on the features $F_t^V$ and $F_t^A$ to produce a predicted image $\hat{I}_{t+1}$.

## 3.3 ACTION EXPERT

**MoE Architecture.** (Black et al.; Intelligence et al., 2025) While our large VLA backbone excels at representation learning, its size is prohibitive for real-time control. To address this, we introduce a lightweight action expert (500M) that operates alongside the main VLA Expert (our full VLA backbone) in a Mixture-of-Experts (MoE) architecture. The Action Expert shares a similar transformer block structure with the VLA Expert but uses a much smaller hidden dimension. This architectural similarity enables a deep and efficient fusion of information via a *Joint Attention* mechanism as Figure 3(a) shows. In this setup, both experts first compute their respective Query, Key, and Value matrices. These matrices are then concatenated along the token sequence dimension to create a single set of inputs for a **Joint Attention** operation

$$Q = [Q_{\text{VLA}}; Q_{\text{AE}}], \quad K = [K_{\text{VLA}}; K_{\text{AE}}], \quad V = [V_{\text{VLA}}; V_{\text{AE}}].$$ (4)

The resulting attention output is then split and routed back to each corresponding expert as Figure 3 shows. This approach allows for a tight, symmetric integration of the VLA's rich representations and the Action Expert's specialized context within a single, efficient computation.

This efficient MoE architecture also serves as an ideal test bed for systematically investigating three distinct action decoding strategies: a **query-based**, an **autoregressive**, and a **flow matching** expert. A key commonality among these variants is the prefilling of the previous action's features ($A_{t-1}$), which provides a strong temporal prior for the current decision.

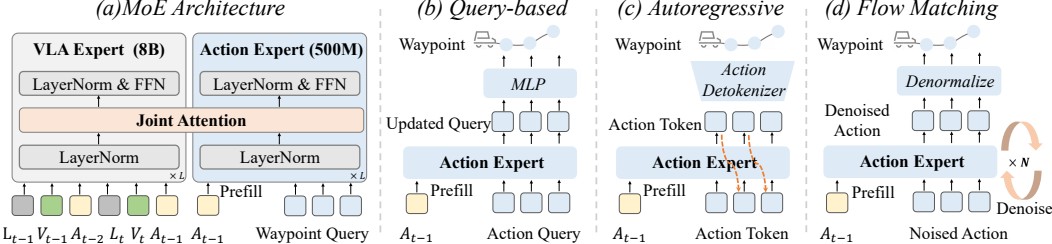

Figure 3: **(a)** Our Mixture-of-Experts (MoE) architecture pairs a large VLA Expert with a lightweight Action Expert for efficient inference. **(b-d)** This framework serves as a testbed for comparing three action decoding schemes: query-based, autoregressive, and flow matching.

**Query-based Action Expert.** This expert employs a set of learnable action queries that interact with the VLA's multimodal context via joint attention. The resulting updated queries are then projected by an MLP head to directly regress the continuous waypoint trajectory. The model is optimized by minimizing the L1 distance between the predicted and ground-truth trajectories.

**Autoregressive Action Expert.** This expert generates actions by autoregressively predicting a sequence of discrete tokens. Its training objective and formulation are identical to those used for our VLA Baseline (described in Section 3.1), minimizing a standard cross-entropy loss.

**Flow Matching Action Expert.** In contrast to the discrete nature of the autoregressive approach, we also implement a continuous action generation method based on flow matching. This method learns a conditional vector field, $v_\phi$, that defines a direct "path" from a simple noise distribution to the complex distribution of real-world driving actions. During training, we define a simple straight-line trajectory between a random noise sample and a ground-truth action (Liu et al., 2022). The model is then optimized via a mean squared error loss to predict a vector field $v_\phi$ that aligns with this trajectory at each point, conditioned on the multimodal context $\mathbf{c}_t$. For inference, we simply start with a new noise sample and follow the learned vector field for a fixed number of steps using a numerical ODE solver. This process deterministically transforms the noise into a precise, continuous action that lies on the learned data manifold.

# 4 EXPERIMENT

## 4.1 DATASETS

**NAVSIM.** We use the NAVSIM (Dauner et al., 2024) benchmark derived from OpenScene (Contributors, 2023), for evaluating performance in safety-critical scenarios.

*NAVSIM v1* metrics include No at-fault Collision (NC), Drivable Area Compliance (DAC), Time-To-Collision (TTC), Comfort (C.), and Ego Progress (EP). NAVSIM uses the Predictive Driver Model Score (PDMS) to evaluate model performance: $PDMS = NC \times DAC \times \frac{5 \times EP + 5 \times TTC + 2 \times C.}{12}$.

*NAVSIM v2* (Cao et al., 2025) includes several components, categorized as penalties or weighted subscores. Key metrics are No at-fault Collision (NC), Drivable Area Compliance (DAC), Driving Direction Compliance (DDC), Traffic Light Compliance (TLC), Ego Progress (EP), Time to Collision (TTC), Lane Keeping (LK), History Comfort (HC), and Extended Comfort (EC). NAVSIM v2 uses the Extended Predictive Driver Model Score (EPDMS) to evaluate model performance: EPDMS = NC $\times$ DAC $\times$ DDC $\times$ TLC $\times \frac{5 \times EP + 5 \times TTC + 2 \times LK + 2 \times HC + 2 \times EC}{16}$.

**In-house Dataset.** To test data scalability beyond academic benchmarks, we use a massive in-house dataset for training and evaluation. Our training set contains *70 million frames* from over *1 million unique clips*. It is curated to be diverse and balanced across a wide spectrum of driving scenarios, while being significantly enriched with challenging and safety-critical events. The test set comprises 100 challenging scenarios. We evaluate trajectory using the Average Displacement Error (ADE) over a 3-second, 6-waypoint future trajectory (2 Hz), and safety using a Collision Rate. We compute our Collision Rate using the same methodology as the No at-fault Collision (NC) metric from the NAVSIM benchmark.

Table 1: **Comparison with state-of-the-art methods on the NAVSIM v1.** NC: no at-fault collision. DAC: drivable area compliance. TTC: time-to-collision. C.: comfort. EP: ego progress. PDMS: the predictive driver model score. Abbreviations: 1x Cam (single front-view camera), Nx Cam (surround-view cameras), L (LiDAR). *: Using the query-based action expert. †: Using the query-based action expert with multiple trajectory anchors following Li et al. (2024b). ‡: Using the AR action expert with the best-of-N (N=6) strategy following Zhou et al. (2025d).

| Method | Ref | Sensors | NC ↑ | DAC ↑ | TTC ↑ | C. ↑ | EP ↑ | PDMS ↑ |
|---|---|---|---|---|---|---|---|---|
| Human | - | - | 100 | 100 | 100 | 99.9 | 87.5 | 94.8 |
| *BEV-based Methods* | | | | | | | | |
| UniAD (Hu et al., 2022) | CVPR'23 | 6x Cam | 97.8 | 91.9 | 92.9 | **100.0** | 78.8 | 83.4 |
| TransFuser (Prakash et al., 2021) | TPAMI'23 | 3x Cam + L | 97.7 | 92.8 | 92.8 | **100.0** | 79.2 | 84.0 |
| PARA-Drive (Weng et al., 2024) | CVPR'24 | 6x Cam | 97.9 | 92.4 | 93.0 | 99.8 | 79.3 | 84.0 |
| LAW (Li et al., 2024a) | ICLR'25 | 1x Cam | 96.4 | 95.4 | 88.7 | 99.9 | 81.7 | 84.6 |
| Hydra-MDP (Li et al., 2024b) | arXiv'24 | 3x Cam + L | 98.3 | 96.0 | 94.6 | **100.0** | 78.7 | 86.5 |
| DiffusionDrive (Liao et al., 2025) | CVPR'25 | 3x Cam + L | 98.2 | 96.2 | 94.7 | **100.0** | 82.2 | 88.1 |
| WoTE (Li et al., 2025a) | ICCV'25 | 3x Cam + L | 98.5 | 96.8 | 94.4 | 99.9 | 81.9 | 88.3 |
| *VLA-based Methods* | | | | | | | | |
| DriveVLA-W0* | - | 1x Cam | 98.7 | 96.2 | 95.5 | **100.0** | 82.2 | 88.4 |
| AutoVLA (Zhou et al., 2025d) | NeurIPS'25 | 3x Cam | 98.4 | 95.6 | **98.0** | 99.9 | 81.9 | 89.1 |
| ReCogDrive (Li et al., 2025b) | arXiv'25 | 3x Cam | 98.2 | 97.8 | 95.2 | 99.8 | 83.5 | 89.6 |
| DriveVLA-W0† | - | 1x Cam | 98.7 | **99.1** | 95.3 | 99.3 | 83.3 | 90.2 |
| AutoVLA† (Zhou et al., 2025d) | NeurIPS'25 | 3x Cam | 99.1 | 97.1 | 97.1 | **100.0** | 87.6 | 92.1 |
| DriveVLA-W0‡ | - | 1x Cam | **99.3** | 97.4 | 97.0 | 99.9 | **88.3** | **93.0** |

Table 2: **Comparison with state-of-the-art methods on the NAVSIM v2 with extended metrics.** NC: No at-fault Collision. DAC: Drivable Area Compliance. DDC: Driving Direction Compliance. TLC: Traffic Light Compliance. EP: Ego Progress. TTC: Time to Collision. LK: Lane Keeping. HC: History Comfort. EC: Extended Comfort. EPDMS: Extended Predictive Driver Model Score.

| Method | NC ↑ | DAC ↑ | DDC ↑ | TLC ↑ | EP ↑ | TTC ↑ | LK ↑ | HC ↑ | EC ↑ | EPDMS ↑ |
|---|---|---|---|---|---|---|---|---|---|---|
| Ego Status | 93.1 | 77.9 | 92.7 | 99.6 | 86.0 | 91.5 | 89.4 | **98.3** | 85.4 | 64.0 |
| TransFuser (Prakash et al., 2021) | 96.9 | 89.9 | 97.8 | 99.7 | 87.1 | 95.4 | 92.7 | **98.3** | 87.2 | 76.7 |
| HydraMDP++ (Li et al., 2024b) | 97.2 | 97.5 | **99.4** | 99.6 | 83.1 | 96.5 | 94.4 | 98.2 | 70.9 | 81.4 |
| DriveSuprem (Yao et al., 2025) | 97.5 | 96.5 | **99.4** | 99.6 | **88.4** | 96.6 | 95.5 | **98.3** | 77.0 | 83.1 |
| ARTEMIS (Feng et al., 2025) | 98.3 | 95.1 | 98.6 | **99.8** | 81.5 | 97.4 | 96.5 | **98.3** | - | 83.1 |
| DiffusionDrive (Liao et al., 2025) | 98.2 | 95.9 | **99.4** | **99.8** | 87.5 | 97.3 | **96.8** | **98.3** | **87.7** | 84.5 |
| DriveVLA-W0 | **98.5** | **99.1** | 98.0 | 99.7 | 86.4 | **98.1** | 93.2 | 97.9 | 58.9 | **86.1** |

## 4.2 IMPLEMENTATION DETAILS

**Two-stage Training Paradigm.** Our model is trained via a two-stage paradigm designed to first learn rich world representations and then specialize in action generation. In the first stage, we pretrain the VLA backbone utilizing the `6VA` sequence configuration. The model is optimized with a joint objective, combining both the world model loss and the action prediction loss. In the second stage, we integrate the model with Action Expert. The VLA backbone now processes a `2VA` input sequence. While we do not freeze the VLA backbone, the model is only supervised by the action loss from the action expert part.

**NAVSIM.** For experiments on the NAVSIM benchmark, models are pretrained on NuPlan (Caesar et al., 2021) for 8k steps and then fine-tuned on NAVSIM for 4k steps, processing 256x144 images. The training is conducted on 8 NVIDIA L20 GPUs with a global batch size of 48. We used the AdamW optimizer with a cosine learning rate schedule, an initial learning rate of $2e - 4$, and bfloat16 mixed-precision. For our ablation studies, we select **DriveVLA-W0 (VQ)** as the default model due to its architectural simplicity.

**In-house Dataset.** For our large-scale experiments on the in-house dataset, models are pretrained for 50k steps and fine-tuned for 30k steps using the same data. This training utilized a cluster of 64 GPUs with a global batch size of 256. The optimizer and learning rate schedule remained identical to the NAVSIM setup.

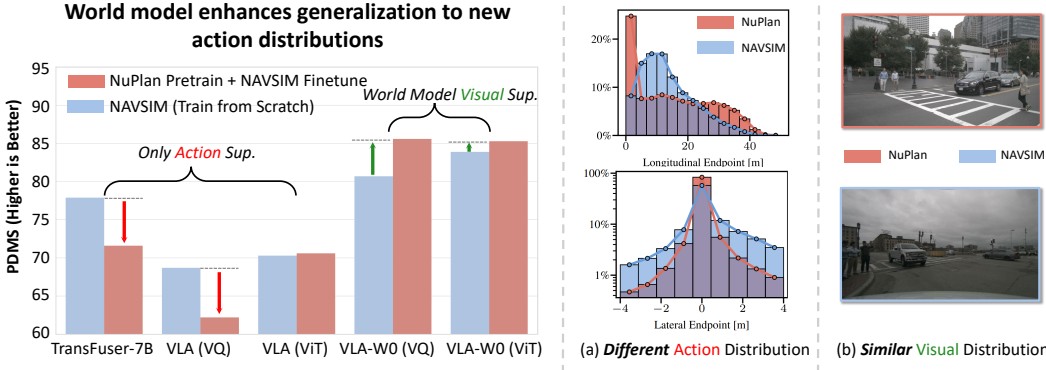

Figure 4: **World modeling unlocks generalization with data scaling.** Our world model turns pretraining from a detriment for sparse-supervision baselines (red arrows) into a benefit for our VLA-W0s (green arrows), enabling positive knowledge transfer across datasets with similar visuals (b) but different action distributions (a). Figure (a) is from Dauner et al. (2024).

**Re-implemented TransFuser.** To provide a solid baseline for our in-house dataset, we adapt and re-implement the well-known TransFuser (Prakash et al., 2021) architecture. For a fair comparison with our single-camera setup, we modify Latent-TransFuser (an image-only variant of TransFuser) to process a single front-view image instead of its original multi-view input. To investigate the impact of model size, we implement two versions: a 50M-parameter model and a 7B-parameter model. The smaller **TransFuser-50M** uses a standard ResNet-34 backbone. The larger **TransFuser-7B** employs a ViT-7B backbone, initialized with pretrained weights from DINOv3 (Siméoni et al., 2025).

### 4.3 COMPARISON WITH STATE-OF-THE-ART METHODS

As presented in Table 1 and Table 2, DriveVLA-W0 establishes a new state-of-the-art on the NAVSIM benchmark by surpassing top-performing methods across different architectural paradigms, including the BEV-based WoTE (Li et al., 2025a) and the VLA-based AutoVLA (Zhou et al., 2025d). Notably, our model achieves this top performance using only a single front-view camera, surpassing competitors that rely on richer sensor suites combining multi-view cameras and LiDAR. This superior performance is attributed to the powerful, dense supervision from our world modeling objective, which enables the model to learn a more effective feature representation.

### 4.4 WORLD MODELS AMPLIFY DATA SCALING LAW

**World modeling unlocks generalization with data scaling.** To test generalization across differing action distributions, we evaluate models by pretraining on the large-scale NuPlan dataset and then fine-tuning on NAVSIM, a smaller benchmark focused on challenging, long-tail maneuvers. This setup creates a significant domain shift in the action space while the visual domain remains similar, posing a challenge for knowledge transfer. As shown in Figure 4 and Table 7, world modeling is crucial for effective transfer. We observe two opposing trends: 1) *Baseline models that rely solely on sparse action supervision often suffer from pretraining.* For instance, TransFuser-7B and our VLA (VQ) baseline show significant performance degradation (red arrows). This is due to overfitting to NuPlan's action distribution, which creates a detrimental prior that hinders adaptation to NAVSIM's corner cases. 2) *Our VLA-W0 models consistently benefit from pretraining.* By forcing the model to predict future visual frames, the world modeling encourages the learning of transferable visual representations of the environment. These transferable visual representations lead to superior generalization.

**World modeling outperforms action-only supervision with data scaling.** We investigate data scalability by training models on three data scales (70k, 700k, and 70M frames). To cleanly ablate the impact of our world modeling paradigm, we conduct these experiments directly on the base VLA models, excluding the action experts. As shown in Table 3, the world model is critical for unlocking the benefits of large-scale data. While baseline models that rely on sparse action supervision quickly

show performance saturation, our DriveVLA-W0 models demonstrate sustained improvement. The impact is most pronounced at the 70M-frame scale. Here, adding world modeling substantially boosts performance, improving the VQ model's ADE by 28.8% and the ViT model's collision rate by 15.9%. This demonstrates that even with a massive volume of training data, action-only supervision cannot replicate the qualitative advantage provided by our dense world model objective.

Table 3: **World modeling outperforms action-only supervision with data scaling.** Unlike baseline models that plateau early under sparse supervision, our VLA-W0 models show consistent improvement.

| Model | In-house Dataset Scale (Number of Frames) | | | | | |
|---|---|---|---|---|---|---|
| | 70k | | 700k | | 70M | |
| | ADE (m) ↓ | Collision (%) ↓ | ADE (m) ↓ | Collision (%) ↓ | ADE (m) ↓ | Collision (%) ↓ |
| TransFuser-50M | 2.5893 | 0.0894 | 1.7464 | 0.0563 | 1.2627 | 0.0472 |
| TransFuser-7B | 2.5757 | 0.0839 | 2.1391 | 0.0710 | 1.2244 | 0.0539 |
| VLA (VQ) (Baseline) | 2.8520 | 0.0982 | 1.5424 | 0.0565 | 1.4829 | 0.0488 |
| + World Model (Ours) | 2.7482 ↑3.6% | 0.0956 ↑2.7% | 1.5985 ↓3.6% | 0.0520 ↑8.0% | 1.0563 ↑28.8% | 0.0392 ↑19.7% |
| VLA (ViT) (Baseline) | 3.1524 | 0.0950 | 1.4202 | 0.0462 | 1.1051 | 0.0359 |
| + World Model (Ours) | 2.5268 ↑19.9% | 0.0834 ↑12.2% | 1.3436 ↑5.4% | 0.0513 ↓11.0% | 1.0640 ↑3.7% | 0.0302 ↑15.9% |

**Action experts reverse performance with data scaling.** Our comparison of action expert on NAVSIM (100k frames) and in-house dataset (70M frames) reveals a striking performance reversal, driven by a trade-off between **prediction precision** and **modeling capacity**. As shown in Table 4, on the small-scale NAVSIM dataset, continuous decoders like query-based and flow matching excel. Here, the trajectory distribution is simple. The higher precision gives these experts an edge over the discrete autoregressive approach, which is hindered by quantization error. However, on our massive dataset, modeling a vastly more complex trajectory distribution becomes the dominant challenge. In this high-data regime, the autoregressive decoder's strong modeling capacity and sample-efficient, teacher-forced training allow it to scale most effectively. Conversely, the query-based expert faces a representational bottleneck, and the flow matching expert proves too sample-inefficient to converge on the complex data manifold, leading to the observed performance reversal.

Table 4: **Action experts reverse performance with data scaling.** For each dataset, all three experts are initialized from an identical pretrained VLA model. Notably, the query-based expert that performs best on the small-scale data is surpassed by the autoregressive expert at the larger scale, highlighting a clear performance reversal.

| Action Expert | NAVSIM (103k Frames) | | | | | | In-house Dataset (70M Frames) | |
|---|---|---|---|---|---|---|---|---|
| | NC ↑ | DAC ↑ | TTC ↑ | C. ↑ | EP ↑ | PDMS ↑ | ADE (m) ↓ | Collision (%) ↓ |
| Query-based | 98.7 | 96.2 | 95.5 | 100.0 | 82.2 | 88.4 | 1.1248 | 0.0453 |
| Flow Matching | 98.4 | 95.3 | 95.2 | 100.0 | 80.9 | 87.2 ↓1.4% | 1.0362 ↑7.9% | 0.0398 ↑12.1% |
| Autoregressive | 98.4 | 93.6 | 94.5 | 100.0 | 79.3 | 85.3 ↓3.6% | 1.0069 ↑10.5% | 0.0295 ↑34.9% |

## 4.5 ABLATION STUDY

The following two world model ablations are conducted without the action experts.

**Vision-Only vs. Vision-Action Sequence** As shown in Table 5, pretraining with an interleaved vision-action sequence (6VA) yields substantial gains over a baseline pretrained with only a vision loss (6V), improving the PDMS score from 84.1 to 85.6. *This demonstrates that grounding visual predictions in corresponding ego actions is crucial.* This conditioning compels the model to learn the environment's underlying causal dynamics, as it must predict the specific visual outcome of an action rather than a generic or ambiguous future.

**Ablation on Sequence Length** We ablate the pretraining sequence length using three configurations: VA, 2VA, and 6VA. Table 6 reveals a clear trend where performance scales with temporal context, with the longest sequence (6VA) achieving the best results. This highlights that a longer context window is critical for learning complex, long-horizon environmental dynamics, as it enables the model to better capture temporal dependencies for planning.

Table 5: **Ablation study on vision-only vs. vision-action sequence design.**

| Pretrain (NuPlan) | Finetune (NAVSIM) | NC ↑ | DAC ↑ | PDMS ↑ |
|---|---|---|---|---|
| / | 2VA | 97.1 | 90.3 | 80.7 |
| 6V | 2VA | 97.9 | 92.8 | 84.1 |
| 6VA | 2VA | 98.3 | 93.8 | 85.6 |

Table 6: **Ablation study on varying the sequence length.**

| Pretrain (NuPlan) | Finetune (NAVSIM) | NC ↑ | DAC ↑ | PDMS ↑ |
|---|---|---|---|---|
| VA | VA | 96.8 | 92.7 | 83.3 |
| 2VA | 2VA | 97.3 | 93.2 | 84.2 |
| 6VA | 2VA | 98.3 | 93.8 | 85.6 |

**Ablation on Latency** We validate the efficiency of MoE architecture by measuring inference latency on an H200 GPU. Compared to the baseline DriveVLA-W0 (**117.8ms** latency, 85.6 PDMS), the addition of our query-based MoE expert significantly cuts the latency to **74.3ms** (just 63.1% of the original) while simultaneously boosting performance to 88.4 PDMS. More analysis is shown in Appendix B.1.

## 4.6 VISUALIZATION

Beyond achieving superior planning performance, DriveVLA-W0 exhibits strong capabilities in action-conditioned future generation, highlighting its potential as a reactive simulator. As illustrated in Figure 5, we explicitly condition the model on a counterfactual "decelerate" trajectory. The resulting frames show the surrounding scene flowing past the vehicle at a noticeably reduced rate compared to the ground truth. This precise alignment between control signals and visual dynamics confirms the model's grounded physical understanding. For further qualitative analysis, including **trajectory visualization and case comparisons** (Figures 7, 8, 9, 10), **future image generation** (Figure 13), and **counterfactual reasoning** (Figure 14), please refer to Appendix C.

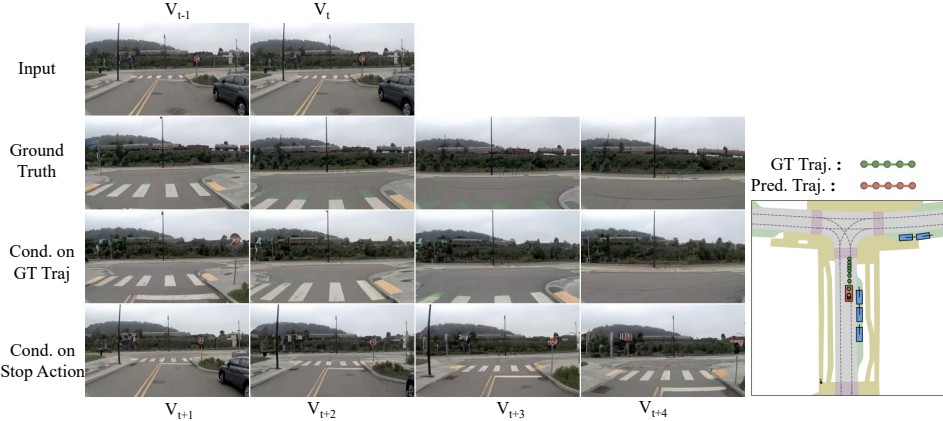

Figure 5: **Action-Conditioned Simulation.** We demonstrate the utility of our world model as a simulator capable of generating diverse futures based on control signals.

## 5 CONCLUSION

In this work, we identified the "supervision deficit" as a fundamental bottleneck hindering the scalability of Vision-Language-Action models in autonomous driving. We proposed DriveVLA-W0, a paradigm that remedies this issue by employing future image prediction as a dense, self-supervised objective, tailored for both VQ- and ViT-based architectures. Our extensive experiments demonstrate that this approach not only unlocks superior data scalability and generalization compared to baselines, but also reveals a compelling performance reversal among action decoders at scale, where simpler autoregressive models ultimately prevail. Ultimately, our findings suggest that embracing dense, predictive world modeling is a crucial step toward realizing the full potential of large-scale data in the pursuit of a more generalized driving intelligence.

## 6 ACKNOWLEDGMENTS

This work was supported in part by Beijing Natural Science Foundation (No. L257015, No. L257004) and in part by the National Natural Science Foundation of China (No. 62320106010).

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

## APPENDIX

## A  OVERALL

In this appendix, we provide supplementary materials to support the main paper. §B presents additional experiments, including a detailed latency analysis, a study on the correlation between generative fidelity and planning performance, and an ablation on the world model's time horizon. We offer further qualitative results in §C, with visualizations of challenging case studies and images generated by our world model. More implementation details for our baselines and training setup are provided in §D. An expanded discussion of related work is available in §E. We disclose our use of LLMs for writing assistance in §F.

## B  MORE EXPERIMENTS

Table 7: **World model enhances generalization to new action distributions.** This table presents the detailed result of Figure 4.

| Model | NAVSIM (Train from Scratch) | | | | | | NuPlan Pretrain + NAVSIM Finetune | | | | | |
|---|---|---|---|---|---|---|---|---|---|---|---|---|
| | NC ↑ | DAC ↑ | TTC ↑ | C. ↑ | EP ↑ | PDMS ↑ | NC ↑ | DAC ↑ | TTC ↑ | C. ↑ | EP ↑ | PDMS ↑ |
| TransFuser-50M | 93.2 | 91.7 | 86.2 | 100.0 | 76.4 | 79.2 | 96.0 | 93.6 | 90.2 | 100.0 | 80.1 | 83.6↑5.5% |
| TransFuser-7B | 95.7 | 89.1 | 89.5 | 100.0 | 73.0 | 77.9 | 93.7 | 84.2 | 86.3 | 100.0 | 67.0 | 71.6↓8.1% |
| VLA-VQ | 93.3 | 80.9 | 86.0 | 100.0 | 63.0 | 68.7 | 90.0 | 76.1 | 82.2 | 100.0 | 56.7 | 62.2↓9.5% |
| VLA-W0-VQ (Ours) | 97.1 | 90.3 | 92.3 | 100.0 | 74.9 | 80.7 | 98.3 | 93.8 | 94.2 | 100.0 | 80.0 | 85.6↑6.1% |
| VLA-ViT | 93.5 | 82.5 | 86.8 | 100.0 | 64.6 | 70.3 | 93.3 | 82.8 | 86.9 | 100.0 | 64.7 | 70.6↑0.4% |
| VLA-W0-ViT (Ours) | 98.0 | 92.6 | 94.2 | 100.0 | 77.7 | 83.9 | 98.3 | 93.5 | 94.8 | 100.0 | 79.1 | 85.3↑1.7% |

### B.1  LATENCY ANALYSIS

We analyze the inference latency of our MoE-based action experts against the full DrivingVLA-W0 backbone, with results shown in Figure 6. In our setup, the flow matching expert is configured to

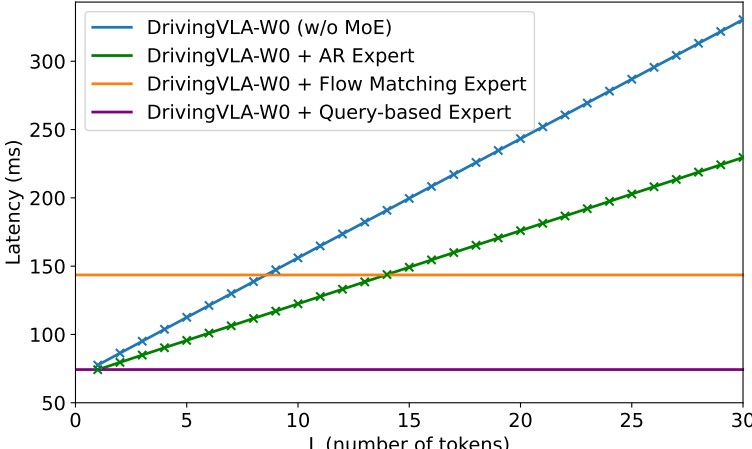

Figure 6: **Latency analysis.** The latency of the AR expert and the VLA baseline scale linearly with the number of generated tokens L, while the flow matching and query-based experts maintain a constant inference time.

use 10 denoising steps, while the query-based expert generates the full trajectory in a single forward pass. The latency of the AR expert, similar to the full VLM backbone, is proportional to the number of generated action tokens L. The results clearly demonstrate that the MoE architecture provides a substantial acceleration. The query-based expert is the most efficient, maintaining a constant low latency of approximately 74ms. The flow matching expert also has a constant latency, albeit higher at around 145ms. To contextualize the AR expert's performance, we consider the average trajectory token length. On the NAVSIM dataset, where trajectories average 5.6 tokens, the AR expert is highly efficient at 95ms, significantly outperforming the full backbone (118ms). For the more complex trajectories in our in-house dataset, which average 17.8 tokens, the AR expert's latency increases to 170ms, but it remains much faster than the baseline's 240ms. This analysis confirms that our MoE framework is critical for achieving the low-latency performance required for real-time deployment.

### B.2 POSITIVE CORRELATION BETWEEN GENERATIVE FIDELITY AND PLANNING PERFORMANCE

To investigate the link between a world model's generative fidelity and its downstream planning performance, we conduct this experiment. First, we establish two models with different generative capabilities by pretraining them on NuPlan with varying context lengths: a 6VA model conditioned on five prior vision-action pairs, and a 2VA model conditioned on just one. As expected, evaluating these checkpoints directly shows that the 6VA model's longer context yields a superior FID score (Table 8), confirming its higher generative fidelity. With these models of varying generative quality established, we

Table 8: **Positive Correlation Between Generative Fidelity and Planning Performance.** *: indicates the images reconstructed by the MoVQGAN encoder-decoder, which serves as an upper bound for generative quality.

| Sequence Design | FID $\downarrow$ | PDMS$\uparrow$ |
|---|---|---|
| 2VA | 9.847 | 84.1 |
| 6VA | 4.610 | 85.6 |
| Upper Bound* | 3.007 | - |

then test their potential for downstream planning. We fine-tune both NuPlan-pretrained checkpoints on NAVSIM under an identical setting (the same as the bottom two rows in Table 5) and evaluate their final PDMS scores. The results confirm a strong positive correlation: the 6VA checkpoint, which had superior generative fidelity, also achieves higher planning performance after fine-tuning. This provides compelling evidence that **the model's ability to generate high-quality, realistic future images is directly linked to its capacity for producing high-quality trajectories.**

### B.3 ABLATION STUDY

**The Time Horizon of World Model** We conduct an ablation study to identify the optimal temporal horizon for our world model's input. Using a shared `6VA`-pretrained checkpoint, we fine-tune and evaluate three configurations that differ in their temporal input range, as shown in Table 9: i) `VA`: Uses only the current visual frame, providing no historical visual context. ii) `VAVA (1s)`: Uses the current frame and a second frame from 1 second in the past. iii) `VAVA (4s)`: Uses the current frame and a second frame from 4 seconds in the past. The `VAVA (1s)` configuration achieves the best overall performance, reaching the highest PDMS score of 85.6. This suggests an optimal trade-off in the temporal input. The `VA` setting, lacking a second visual frame, struggles to capture the environment's dynamic information. Conversely, the 4-second interval in the `VAVA (4s)` setting introduces excessive scene variation between the two distant frames, which likely makes the future prediction task more challenging for the model.

## C VISUALIZATION

### C.1 PLANNING TRAJECTORY

In this section, we provide a qualitative case analysis to offer deeper insights into our model's behavior and validate our key findings. Our analysis is twofold. First, as shown in Figure 7, we compare our DriveVLA-W0 against baseline models (VLA baseline and TransFuser) in a challenging corner case to visually demonstrate the benefits of world modeling for navigating complex scenarios. Second, we compare the trajectories generated by different action experts built upon the same VLA backbone. The subsequent visualizations in Figure 8,9,10 reveal that **the AR expert generates markedly more stable trajectories than the flow-matching approach**, particularly in terms of inter-frame consistency and avoiding aggressive maneuvers.

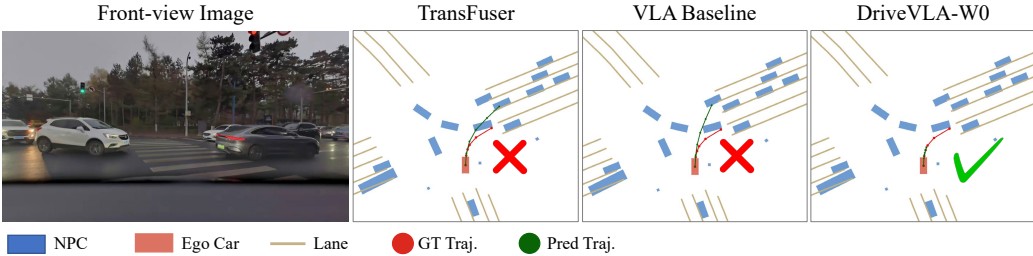

Figure 7: **World modeling improves trajectory planning in complex scenarios.** TransFuser and VLA baseline often fail in such interaction scenarios due to their weak ability to predict scene dynamics. However, with the support of world modeling, VLA-W0 possesses strong predictive capabilities, thereby avoiding collision in this type of scenario.

Table 9: **Ablation on the temporal interval for world model inputs.** The results reveal a clear performance trade-off, with a 1-second interval being optimal. Using only the current frame (VA) lacks sufficient temporal context, while a 4-second interval introduces excessive scene variation.

| Pretrain Type | Finetune Type | TI | NC ↑ | DAC ↑ | TTC ↑ | Comf. ↑ | EP ↑ | PDMS ↑ |
|---|---|---|---|---|---|---|---|---|
| 6VA | VA | / | 96.6 | 92.6 | 91.2 | 100.0 | 78.8 | 82.9 |
| 6VA | 2VA | 4s | 97.9 | 93.2 | 93.9 | 100.0 | 78.3 | 84.3 |
| 6VA | 2VA | 1s | 98.3 | 93.8 | 94.2 | 100.0 | 80.0 | 85.6 |

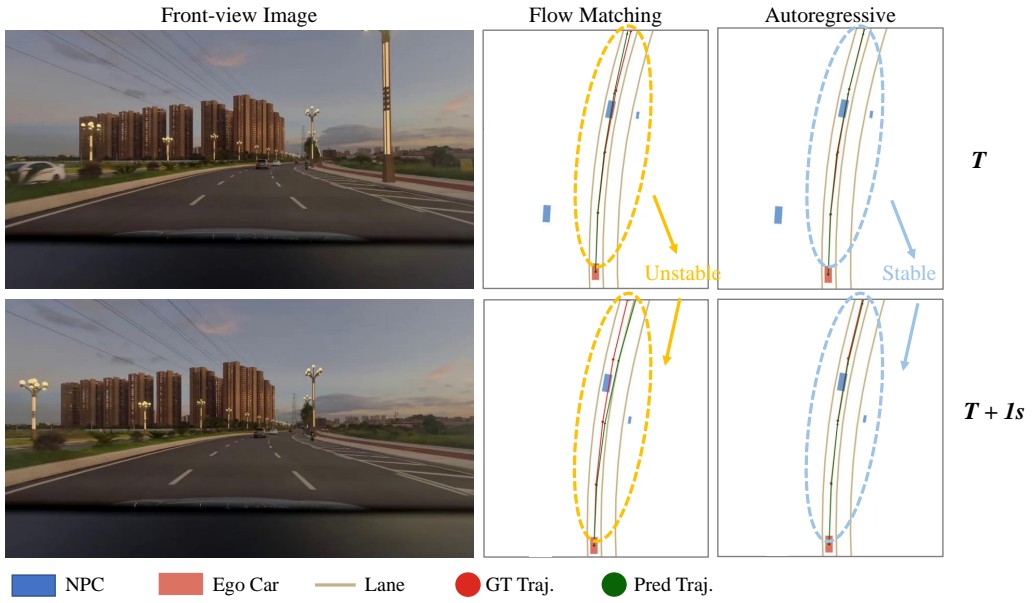

Figure 8: **Comparison of flow matching and autoregressive action expert.** The trajectory generated by the flow matching action expert is relatively **unstable**, with jumps occurring between adjacent frames, while the AR action expert generates much more **stable** trajectories.

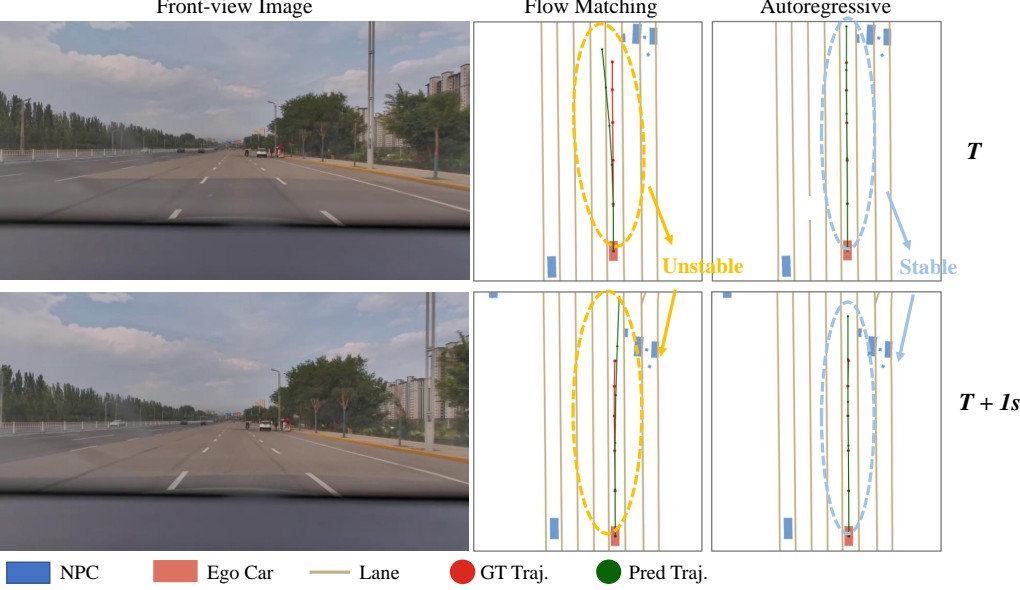

Figure 9: **Comparison of flow matching and autoregressive action expert.** The trajectory generated by the flow matching action expert is relatively **unstable**, with jumps occurring between adjacent frames, while the AR action expert generates much more **stable** trajectories.

## C.2 FAILURE CASE

To provide a balanced view of our model's performance and identify directions for future improvement, we analyze representative failure cases encountered during evaluation. We categorize these failures into two primary types: **instruction ambiguity** and **dynamic prediction errors**. As illustrated in Figure 11, coarse-grained navigation commands can lead to indecision in complex road topologies

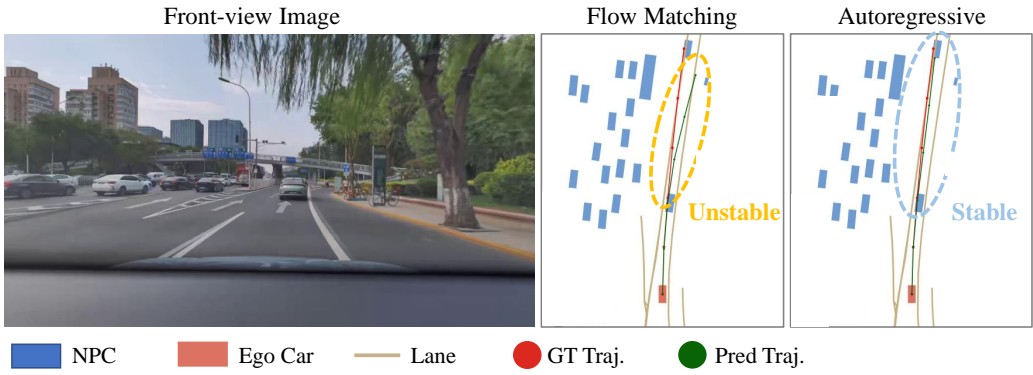

Figure 10: **Comparison of flow matching and autoregressive action expert.** The trajectory generated by the flow matching action expert is relatively unstable, and it may go beyond the drivable area. The AR action expert generates much more stable trajectories.

(e.g., Y-junctions). Additionally, Figure 12 highlights the challenge of predicting fine-grained dynamic objects (e.g., oncoming vehicles) in high-interaction scenarios.

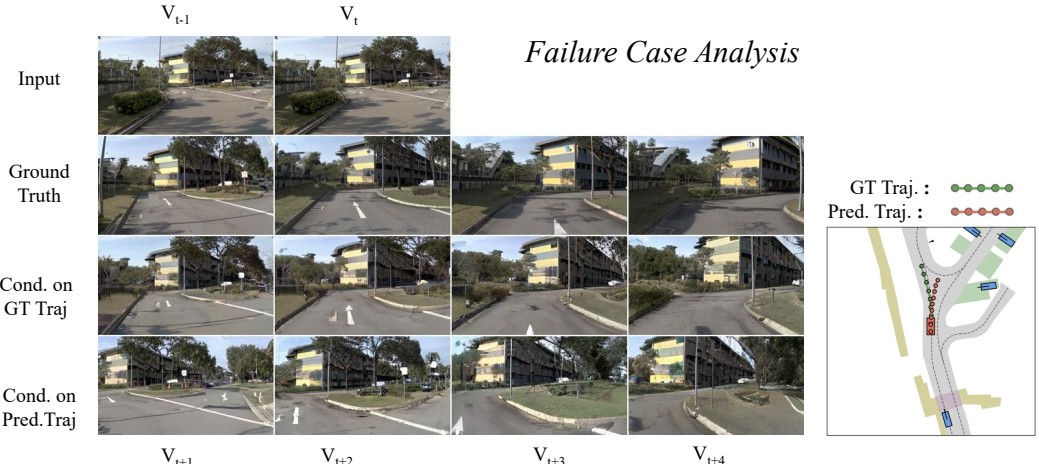

Figure 11: **Failure case analysis: ambiguity in driving commands.** We illustrate a failure mode stemming from the ambiguity of coarse-grained navigation instructions. In this Y-shaped intersection, the model receives a generic "go straight" command. Due to the splitting road topology, this instruction is ambiguous, as it maps clearly to neither the left nor the right branch. Consequently, the model fails to commit to a valid lane, hesitantly driving straight into the fork area (as shown by the predicted red trajectory). This highlights a limitation in the current NAVSIM benchmark, where the discrete command set (restricted to simple Turn Left/Straight/Right) lacks the granularity to resolve ambiguities in complex geometries. **Note:** To faithfully represent the visual information available to the model, all displayed images are *reconstructions from the MoVQGAN tokens*, rather than raw RGB frames.

## C.3 FUTURE IMAGE GENERATION

Our world model demonstrates both **high visual fidelity** and **strong action consistency**. As shown in Figure 13, it generates realistic and plausible images across diverse and challenging scenarios. More importantly, these visual predictions are tightly coupled with the model's planning process.

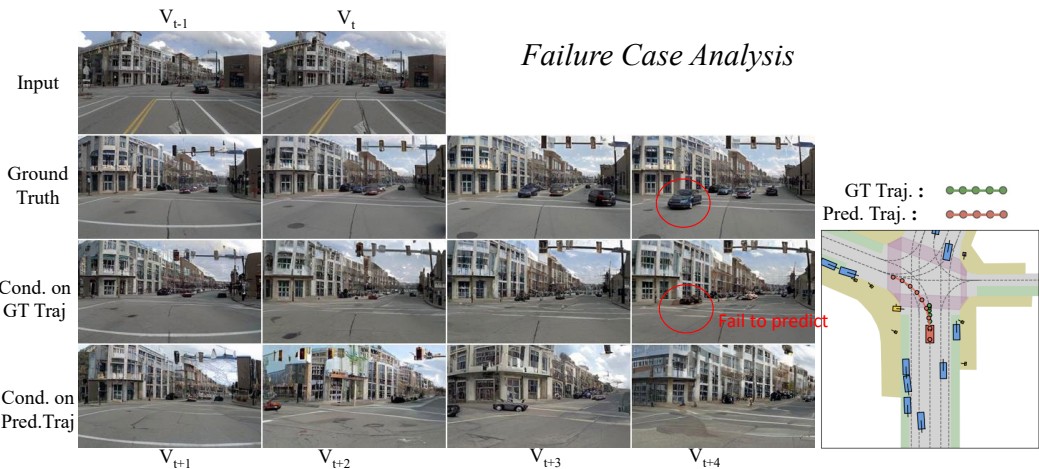

Figure 12: **Failure case analysis: dynamic object prediction.** While our world model generalizes well to most dynamic scenes, it faces challenges in *predicting multiple objects at complex intersections*. In this specific case, the world model fails to anticipate the emergence of oncoming vehicles in the future frames (as reflected in the Cond. on GT Traj). Consequently, the planner, unaware of the potential conflict, incorrectly executes a left turn. This highlights the modeling of fine-grained dynamic objects as a valuable direction for future research. **Note:** To faithfully represent the visual information available to the model, all displayed images are *reconstructions from the MoVQGAN tokens*, rather than raw RGB frames.

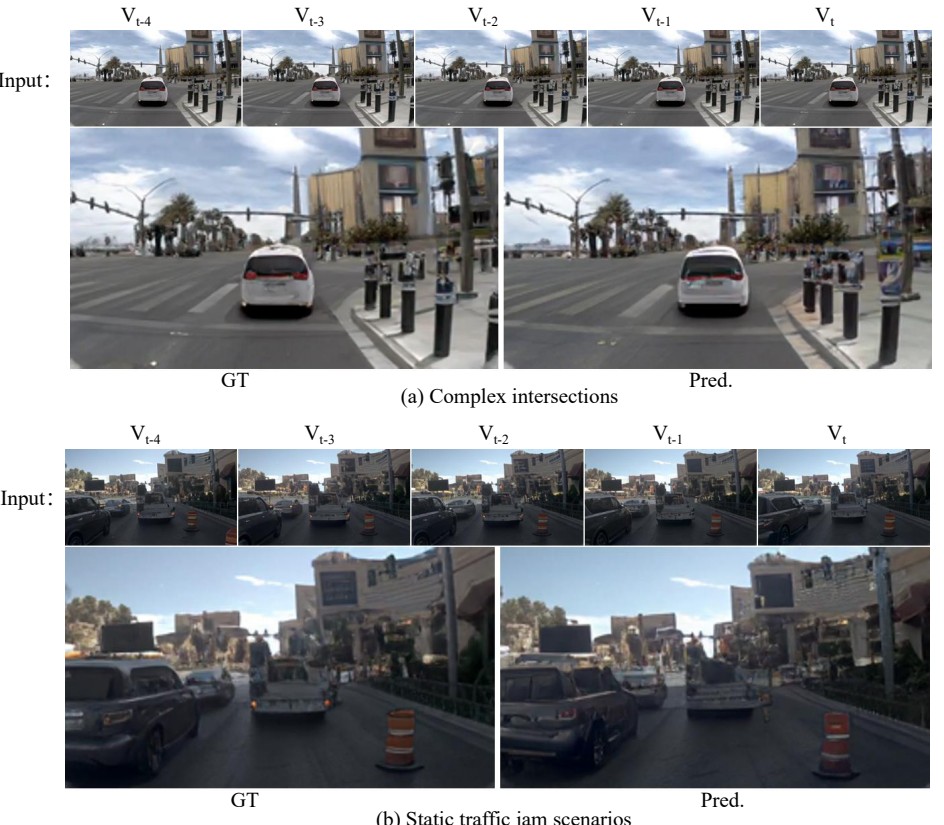

Figure 13: **Future image generation.** Our world model demonstrates strong generative fidelity, producing visually realistic and contextually plausible futures across diverse and challenging scenarios, including complex intersections and dense traffic.

### C.4 COUNTERFACTUAL REASONING

This section investigates the model's capability for counterfactual reasoning, synthesizing alternative futures based on hypothetical actions. As shown in Figure 14, conditioning the model on a "turn right" action generates realistic off-road imagery despite the ground truth being straight. This confirms that the model captures the underlying scene geometry rather than merely memorizing training trajectories.

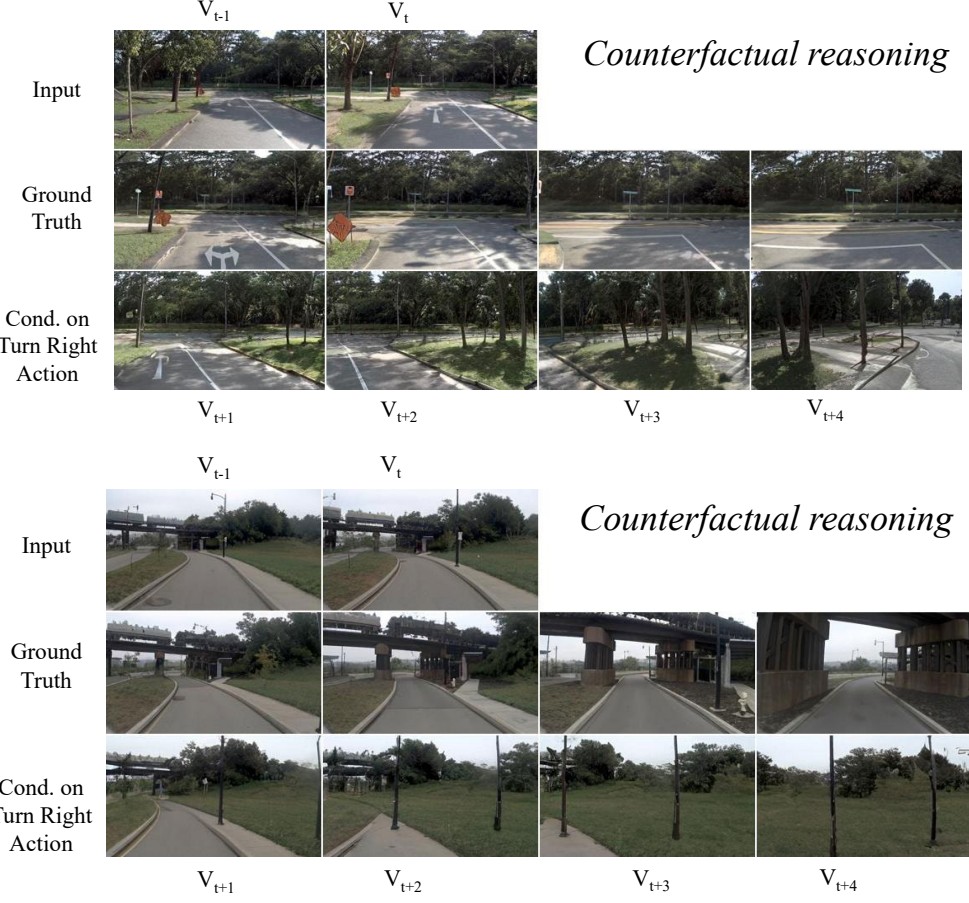

Figure 14: **Counterfactual reasoning.** By conditioning on a counterfactual "turn right" action at the roadside, our world model generates realistic video sequences where the vehicle deviates from the ground truth trajectory and drives off-road. This validates that the model has learned the underlying scene geometry rather than simply memorizing the training data.

## D MORE IMPLEMENTATION DETAILS

**TransFuser** For consistency of the setting, we adopt *Latent TransFuser*, which replaces the LiDAR branch's real sensor features with BEV latent queries. The camera branch supplies the primary semantic and appearance features, while the latent BEV branch provides learnable global query anchors; the two are aligned and made complementary via cross-modal cross-attention. In this way, Latent TransFuser retains the global geometric priors of the BEV view and achieves stable fusion and planning performance under a vision-only setup. We instantiate two image backbones to highlight capacity trade-offs: a lightweight *ResNet-34* variant (*TransFuser-50MB*) and a large *DINOv3 ViT-7B* variant (*TransFuser-7B*). Concretely, the inputs are single front-view RGB image $\mathbf{I}$ and a latent BEV query tensor $\mathbf{Q}_{\text{bev}}$ of size $H \times W$. After an image encoder (ResNet-34 or DINOv3) and a BEV latent encoder, we obtain $\mathbf{F}_{\text{img}}$ and $\mathbf{F}_{\text{bev}}$, which are then aligned and fused through a multi-scale

cross-attention module to produce the fused representation $\mathbf{F}_{\text{fuse}}$. A planning head predicts $K$ future waypoints $\{\hat{\mathbf{p}}_{t+1}, \ldots, \hat{\mathbf{p}}_{t+K}\}$ based on $\mathbf{F}_{\text{fuse}}$.

## E   MORE RELATED WORK

**Scaling Laws in Deep Learning.** Kaplan et al. (2020) were the first to systematically demonstrate that pretraining loss scales as a power law with model size, dataset size, and computational cost. Chinchilla (Hoffmann et al., 2022) then showed many LMs were undertrained and derived a compute-optimal prescription that scales model size and tokens proportionally. In computer vision, Zhai et al. (2022) charted ViT scaling law with stable training recipes, and ViT-22B (Dehghani et al., 2023) scaled ViTs to 22B parameters, verifying predictable multi-task improvements. Lin et al. (2025) conducted a large-scale study of imitation-learning data scaling in robotics, and found near–power-law gains from increasing environmental and object diversity with improved zero-shot generalization. In autonomous driving, STR (Sun et al., 2023) shows large trajectory models scale steadily in both prediction and planning, and Baniodeh et al. (2025) reported power-law improvements for joint motion forecasting and planning with large driving datasets. For end-to-end driving, Naumann et al. (2025) observe roughly log-linear gains in both open- and closed-loop metrics as training data scale increases. Zheng et al. (2024b) also observe power-law improvements from data scaling in a large-scale imitation learning study. However, Zheng et al. (2024b) exclusively analyzes scaling behavior within the conventional paradigm of sparse action supervision. Our work, in contrast, provides the first study of how these scaling laws are reshaped by the self-supervised signals provided by world modeling.

## F   USE OF LLMS

Large Language Models (LLMs) are employed to polish the writing in this manuscript.

