# OpenReview forum: "DriveVLA-W0: World Models Amplify Data Scaling Law in Autonomous Driving"
_ICLR.cc/2026/Conference — ICLR 2026 Poster_

### Official Review · Reviewer_dcaL · 2025-10-28

**Soundness:** 3
**Presentation:** 3
**Contribution:** 3
**Rating:** 6
**Confidence:** 4

**Summary:**

This paper introduces DriveVLA-W0, a training paradigm for VLA models in autonomous driving that addresses the “supervision deficit” in current VLA systems, wherein large model capacity is supervised by sparse, low-dimensional action labels. DriveVLA-W0 supplements action supervision with a dense, self-supervised world modeling objective and demonstrates its method with both autoregressive and diffusion approaches. The authors conduct extensive experiments on the NAVSIM benchmark and a large-scale in-house dataset (70M frames), showing that world modeling not only accelerates performance scaling with data size but also improves generalization, efficiency, and real-time deployment feasibility.

**Strengths:**

- Clearly define the supervision deficit issue and explain why relying solely on action objectives does not scale.
- Introduce rigorous and balanced designs for both discrete (AR/VQ) and continuous (diffusion/ViT) visual representations, with empirical evidence showing generalization benefits under distribution shift and the critical role of dense supervision on data scaling.
- Extensive experiments along with qualitative visualizations that link generative fidelity to planning consistency, reinforcing the claim that predictive world modeling yields richer representations.

**Weaknesses:**

- The claim that “the new training paradigm that uses world modeling as a powerful form of self-supervision to supplement the sparse action signal” is somewhat overclaimed, given that recent methods such as WorldVLA, DOE-1, and LAW similarly adopt self-supervised world modeling to mitigate the sparse action signal.
- Parity of baselines on the in-house dataset is not fully convincing. The re-implementation of TransFuser differs notably in terms of pretraining compared to other approaches, and it typically uses multi-sensor input. Moreover, including more VLM-based methods would help establish a fairer comparison.
- The experimental evaluations are conducted only in a semi-closed-loop, non-reactive setting. It would be beneficial to provide further exploration and demonstration under a fully closed-loop evaluation.

**Questions:**

- In Table 2, why does equipping the world model result in worse performance on the 700k dataset compared to the baseline?
- Can this framework be extended to incorporate multi-view image inputs?
- Regarding latency, is the reported delay solely for the action output, or does it represent the total delay? Could you provide latency metrics with varying sequence lengths (e.g., 2VA, 3VA, etc.)?

---

> ### Author Response · Authors · 2025-11-24
> **Response to Reviewer dcaL (Part1)**
>
> We express our sincere gratitude to Reviewer dcaL for the encouraging assessment and the support of our work. We are particularly gratified that you recognized our rigorous design and the established link between generative fidelity and planning. We also appreciate your constructive feedback on baselines and evaluation settings, which has helped us strengthen the manuscript. Please find our detailed point-by-point responses below.
>
> ### W1
>
> Concerns regarding incremental contribution.
>
> ### R1
>
> We appreciate this feedback and agree that our original phrasing regarding novelty was somewhat overstated. Accordingly, we have revised the **Introduction** and **Related Work** to explicitly discuss these precedents (WorldVLA, DOE-1, LAW) and clarify the specific scope of our contribution. Please refer to the **General Response** for a detailed comparative analysis distinguishing our approach.
>
> ### W2
>
> Inclusion of VLM baselines
>
> ### R2
>
> We appreciate this suggestion. To address this, we have incorporated **ReCogDrive** [1] as an additional strong VLA baseline on our in-house dataset. To ensure a fair comparison, we followed the two stages of their protocol:
>
>  **Stage 1 (VQA pretraining):** We directly utilized the official checkpoints (InternVL3-8B) provided by the ReCogDrive repository, which have been pretrained on massive VQA datasets.
>
> **Stage 2 (Planning fine-tuning):** We loaded this checkpoint and fine-tuned the model on our in-house driving dataset.
> Due to the limited rebuttal period, we conducted experiments at two representative scales: **70k** (small-scale) and **70M** (large-scale). The results are summarized below.
>
> | Method  | VQA Pretraining | World Modeling |  Dataset Scale | ADE | Collision |
> | --- | --- | --- | --- | --- | --- |
> | ReCogDrive [1] | √ | - | 70k | 2.6832 | 0.0863 |
> | DriveVLA-W0 (ViT) | - | - | 70k | 3.1524 | 0.0950 |
> | DriveVLA-W0 (ViT) | - | √ | 70k | 2.5268 | 0.0834 |
> | ReCogDrive [1] | √ | - | 70M | 1.0937 | 0.0368 |
> | DriveVLA-W0 (ViT) | - | - | 70M | 1.1051 | 0.0359 |
> | DriveVLA-W0 (ViT) | - | √ | 70M | 1.0640 | 0.0302 |
>
> **Analysis:**
> These comparisons reveal two critical insights regarding data scaling:
>
> 1. **Small-scale (70k): VQA pretraining aids initialization.** ReCogDrive (ADE 2.68m) significantly outperforms the baseline (ADE 3.15m) and is comparable to our method (ADE 2.52m), confirming the utility of general VQA priors when domain-specific data is scarce.
> 2. **Large-scale (70M): World modeling surpasses static VQA pretraining.** At scale, the advantage of static VQA pretraining diminishes. DriveVLA-W0 clearly outperforms ReCogDrive (Collision 0.0302 vs. 0.0368), demonstrating that world modeling provides a scalable self-supervised signal that amplifies scaling laws more effectively than fixed pretraining.
>
> ### W3
>
> Request for closed-loop evaluation.
>
> ### R3
>
> We appreciate this valuable suggestion to strengthen our evaluation. We expand our experiments as follows:
>
> - **Public Benchmark:** We evaluated our model on **NAVSIM v2**, which introduces reactive traffic agents and a two-stage pseudo-closed-loop simulation. As shown in the table 2 of the revised paper and the table below, DriveVLA-W0 achieves state-of-the-art performance with an **EPDMS of 86.1**, significantly outperforming strong end-to-end baselines like DiffusionDrive (84.5) and HydraMDP++ (81.4).
> - **In-House Simulation:** In our internal closed-loop simulator, we achieved a **Miles Per Intervention (MPI) exceeding 3km**, demonstrating robust long-term driving capability. Notably, this performance is achieved using our **ViT-based model**. Empirically, we found the ViT architecture superior to the VQ-based counterpart in handling fine-grained semantic details, such as traffic lights and small objects, which are critical for closed-loop stability.
> - **Real-World Testing:** We are actively preparing a demonstration of real-vehicle deployment. In the future version, we will upload relevant videos to provide a deeper exploration and visualization of the model's closed-loop performance in complex real-world scenarios.
>
> | **Method** | **NC ↑** | **DAC ↑** | **TTC ↑** | **EPDMS ↑** |
> | --- | --- | --- | --- | --- |
> | HydraMDP++ [2] | 97.2 | 97.5 | 96.5 | 81.4 |
> | DiffusionDrive [3] | 98.2 | 95.9 | 97.3 | 84.5 |
> | **DriveVLA-W0** | **98.5** | **99.1** | **98.1** | **86.1** |
>
> [1]: ReCogDrive: A Reinforced Cognitive Framework for End-to-End Autonomous Driving
>
> [2]: Hydra-MDP++: Advancing End-to-End Driving via Hydra-Distillation with Expert-Guided Decision Analysis
>
> [3]: Diffusiondrive: Truncated Diffusion Model for End-to-end Autonomous Driving

---

> ### Author Response · Authors · 2025-11-24
> **Response to Reviewer dcaL (Part2)**
>
> ### Q1
>
> Performance anomaly at 700k Scale
>
> ### R1
>
> Thank you for this keen observation. Upon rigorous investigation, we identified that the performance dip was caused by **suboptimal convergence** due to an **insufficient training schedule**. In our initial experiments, we scaled training steps linearly (10k, 15k, 30k steps for 70k, 700k, 70M frames respectively), but the world modeling objective requires more iterations to learn effective visual representations. To address this, we increased the training steps for the 700k dataset from **15k** to **20k**. As shown in the updated results below, with adequate training, the model equipped with world modeling now **surpasses** the action-prediction-only baseline in both trajectory accuracy (ADE) and safety (Collision Rate).
>
> | Method | World Model | ADE | Collision |
> | --- | --- | --- | --- |
> | DriveVLA-W0 (ViT) | w/o | 1.3618 | 0.0481 |
> | DriveVLA-W0 (ViT) | w/ | 1.3105 | 0.0423 |
>
> ### Q2
>
> Extension to multi-view image inputs.
>
> ### R2
>
> Yes, our framework is inherently extensible. We demonstrated this by adapting our ViT-based architecture to accept multi-view inputs (front, front-left, front-right, and rear). Specifically, we modified the input sequence to process these four images sequentially. Validated on both NAVSIM and our in-house dataset, this configuration consistently outperforms the single-view baseline (see table below). Notably, it reduces the in-house ADE from **1.0640m** to **0.9395m** and improves NAVSIM PDMS to **84.7**. We attribute these gains to the expanded field of view, .which significantly enhances safety during lane changes. Furthermore, we observe that **world modeling continues to provide substantial gains** in this multi-view setting (boosting PDMS from 69.0 to 84.7), confirming the scalability of our method.
>
> | #View | World Modeling | NC | DAC | PDMS |
> | --- | --- | --- | --- | --- |
> | Single | w/o | 93.5 | 82.5 | 70.3 |
> | Single | w/ | 98.0 | 92.6 | 83.9 |
> | Multi | w/o | 93.9 | 79.6 | 69.0 |
> | Multi | w/ | 98.3 | 93.0 | 84.7 |
>
> | #View | ADE | Collision |
> | --- | --- | --- |
> | Single  | 1.0640 | 0.0302 |
> | Multi | 0.9395 | 0.0282 |
>
> ### Q3
>
> Explanation of latency.
>
> ### R3
>
> Thank you for this valuable question. We clarify that our inference pipeline bypasses image generation to output **only actions**. As a result, the reported metric represents the **total end-to-end latency**. To clearly address your query, we divide our response into two parts: first detailing the specific inference protocol, and then presenting the latency analysis for varying sequence lengths.
>
> 1. Inference Protocol:
> **We clarify that during inference, the model is configured to bypass image generation and solely output actions.** The model inputs historical context (e.g., $I_{t-1}, A_{t-1}, I_t$) and predicts the full 4-second planning trajectory ($A_t$) in a single pass. Generation is terminated immediately after the action tokens are produced. This design is distinct from prior autoregressive paradigms that alternate between generating intermediate future frames and short-horizon actions (e.g., generating a 1s image to predict the next 1s action). By avoiding this computationally expensive loop and predicting the full trajectory directly, we significantly reduce inference latency.
> 2. Latency vs. Sequence Length:
> **We evaluate the latency impact of increasing the historical input window from 2VA to 3VA.** As shown in the table below, the total latency increases from 74.3ms (2VA) to 108.6ms (3VA). The breakdown reveals distinct scaling behaviors: the VLA Backbone's latency scales linearly with context length (increasing $\approx 1.5\times$), while the Action Expert's latency increases only marginally (from 8.0ms to 9.1ms). This efficiency arises because the Expert's FFN computation remains constant for fixed action queries, with only the Joint Attention mechanism slightly affected by the increased context size.
>
> |Setting  | Vision Encoder (MoVQGAN) | VLA Expert (Emu3-8B) | Action Expert (Query-based) | Total Latency|
> | --- | --- | --- | --- | --- |
> | 2VA | 13.7ms | 52.6ms | 8.0ms | 74.3ms |
> | 3VA | 20.6ms | 78.9ms | 9.1ms | 108.6ms |

---

### Official Review · Reviewer_uuy7 · 2025-10-29

**Soundness:** 4
**Presentation:** 3
**Contribution:** 4
**Rating:** 8
**Confidence:** 5

**Summary:**

This paper identifies a critical "supervision deficit" in scaling Vision-Language-Action (VLA) models for autonomous driving, where large-capacity models are underutilized due to sparse action supervision. To address this, the authors propose DriveVLA-W0, a training paradigm that incorporates world modeling as a dense, self-supervised learning signal. This approach is instantiated for both discrete (VQ-based) and continuous (ViT-based) VLA architectures using autoregressive and diffusion world models, respectively. The core contribution is the extensive experimental validation, particularly on a massive 70M-frame dataset, demonstrating that this dense supervision not only improves performance but fundamentally amplifies the data scaling law. The paper also introduces a lightweight MoE-based "Action Expert" to ensure real-time inference.

**Strengths:**

- The paper provides a clear motivation by identifying the "supervision deficit" as a key bottleneck. This framing is insightful and pinpoints a significant challenge in scaling VLA models.

- The empirical evidence is a major strength. The authors validate their approach across two distinct VLA architectures and, most importantly, across vastly different data scales (NAVSIM and a 70M-frame in-house dataset). This large-scale study provides strong evidence for their claims.

- The central claim that world modeling amplifies data scaling, rather than just providing a performance boost, can be a highly impactful finding for the driving community.

- Achieving state-of-the-art results on the NAVSIM benchmark further underscores the effectiveness of the proposed representation learning paradigm.

**Weaknesses:**

- Training Cost Discussion: The addition of a world modeling objective (especially a diffusion-based one) presumably adds significant computational overhead to the training process. A brief discussion on the trade-offs in terms of training cost (e.g., total GPU hours vs. the baseline) would make the analysis more complete.

- Generalization to Other Sensors: The work impressively achieves SOTA with a single camera. However, it would be beneficial to discuss the authors' perspective on how this world modeling approach might apply to richer sensor suites (e.g., surround-view cameras or LiDAR). Would they expect a similar "amplification" effect when the model already has access to more complete 3D spatial information?

**Questions:**

The finding in Table 3 regarding the "performance reversal" of action experts is one of the most interesting parts of the paper. The autoregressive expert, which performs worst on the small-scale NAVSIM dataset, becomes the best-performing model at the 70M-frame scale. Could the authors elaborate on their hypothesis for this reversal? Is this purely a matter of sample efficiency, where the flow-matching expert simply requires even more data to converge on the complex trajectory manifold? Or, do the authors believe that the discrete, tokenized action space of the autoregressive model provides a fundamentally better inductive bias for modeling the highly complex and multi-modal action distributions present in a massive, diverse dataset?

---

> ### Author Response · Authors · 2025-11-24
> **Response to Reviewer uuy7**
>
> We express our sincere gratitude to Reviewer uuy7 for the positive assessment and strong endorsement, particularly for recognizing the significance of the "supervision deficit" and the high impact of our data scaling findings. We also value your insightful feedback on training overhead and sensor generalization, which has helped us strengthen the manuscript. Please find our detailed point-by-point responses below.
>
> ### W1
>
> Training Cost.
>
> ### R1
>
> We thank the reviewer for this valuable suggestion regarding training efficiency. We have conducted a detailed cost analysis, summarized in the table below:
>
> | Model Type  | World model | Training Time (hours) |
> | --- | --- | --- |
> | Qwen2.5-VL | w/ | 4 |
> |  | w/o | 3 |
> | Emu3 | w/ | 13.5 |
> |  | w/o | 13 |
>
> **Analysis:**
>
> - **Emu3 (Unified Architecture):** The overhead is **negligible (+0.5h)**. Since Emu3 is natively designed for next-token generation (visual tokens), adding the world modeling objective does not require additional modules, but simply involves enabling the visual generation loss.
> - **Qwen (VLA + Diffusion Head):** The overhead is more noticeable **(+1.0h)** because this architecture necessitates a separate diffusion decoder for image synthesis. However, despite this addition, the total training time remains highly efficient (4h).
>
> Note: the faster overall speed of Qwen compared to Emu3 is attributed to the mature optimization of the standard LLM ecosystem.
>
> ### W2
>
> Generalization to multi-view sensors.
>
> ### R2
>
> We appreciate this insightful question. In response, we have extended our framework to support **multi-view image inputs** (surround-view cameras) and observed that the "amplification" effect indeed persists, leading to further performance gains. For detailed quantitative results and specific discussions on the multi-view experiments, please kindly refer to our **Response to Reviewer dcaL (Q2)**.
>
> ### Q1
>
> Hypothesis for "Performance Reversal”.
>
> ### R1
>
> This is a profound question. We hypothesize that the primary factor for this reversal is the **convergence efficiency** (or sample efficiency). To verify this, we conducted an ablation study on the 70M-frame dataset. Specifically, we took the VLA backbone pretrained in Stage 1 (without the Action Expert) and fine-tuned it with different Action Experts in Stage 2, monitoring their performance at different training steps (5k vs. 30k). The results are summarized below.
>
> | Action Expert | Training Steps | ADE | Collision |
> | --- | --- | --- | --- |
> | AR  | 5k | 1.0069 | 0.0295 |
> | AR | 30k | 1.0062 | 0.0298 |
> | Flow Matching | 5k | 1.5732 | 0.0574 |
> | Flow Matching | 30k | 1.0362 | 0.0398 |
>
> The results isolate convergence efficiency as the primary factor. The **AR**  expert demonstrates high sample efficiency, achieving near-optimal performance (ADE $\approx$ 1.00) at just 5k steps with negligible gains at 30k. In contrast, **Flow Matching** struggles to converge on this massive manifold. Its performance is poor at 5k (ADE 1.57) and, despite significant recovery by 30k (ADE 1.03), it still fails to match the AR expert.

---

> > ### Comment · Reviewer_uuy7 · 2025-11-27
> >
> > Thank the authors for their detailed response and the additional experiments. The analysis of convergence efficiency effectively resolves my questions regarding the performance reversal between AR and Flow Matching experts, and the results extending to multi-view settings are promising. Given these clarifications and the empirical evidence, I will maintain my recommendation for acceptance.

---

### Official Review · Reviewer_83sR · 2025-11-04

**Soundness:** 3
**Presentation:** 3
**Contribution:** 2
**Rating:** 4
**Confidence:** 5

**Summary:**

This paper introduces DriveVLA-W0, a method that utilizes future image prediction as an auxiliary training objective in addition to the action prediction objective. The paper investigates this method on both VQ- and ViT-based architectures. Experimental results on NAVSIM and an in-house dataset demonstrate the effectiveness of this approach compared to VLAs that only predict actions.

**Strengths:**

S1) The proposed method is well-motivated and writing is easy to follow.

S2) The analysis of why employing next image prediction is beneficial, as also shown in the results of Figure 4, indicates that jointly predicting the next frame leads to more transferable features and helps avoid overfitting to dataset-specific action patterns.

S3) In addition to NAVSIM, the paper also investigates the scalability of their method on a larger in-house dataset.

**Weaknesses:**

W1) Even though the results look good, the paradigm that jointly predicts action and future states (images) is not a new concept, both in autonomous driving (DrivingGPT, Doe-1, VaVAM) and robot learning.

W2) Efficiency concern. The paper investigates both VQ and ViT based architecture. For VQ, the model is trained in an interleaved manner: image, action, image, action. The paper mentions that image generation can be bypassed during inference. However, this may lead to train-infer gap. Otherwise, the latency should be huge.

W3) From Line 316-321, the model is trained in two-stage manner. In the first stage, the model is trained to predict both image and action. However, in the second stage, both VLA backbone and action expert is trained. But why only predict action in this stage. As the author already claimed that jointly predict image and action leads to better representation, will this undermine VLA's representation in the second stage?

**Questions:**

Please check the Weaknesses

---

> ### Author Response · Authors · 2025-11-24
> **Response to Reviewer 83sR**
>
> We sincerely thank Reviewer 83sR for the comprehensive review and the positive recognition of our research motivation and experimental results. We also deeply appreciate your constructive suggestions, which have helped us to improve the quality of our manuscript. Please find our detailed point-by-point responses below.
>
> ### W1
>
> Concerns regarding incremental contribution and prior works.
>
> ### R1
>
> We sincerely thank the reviewer for pointing out these relevant prior works. We have conducted a detailed comparative analysis against the cited methods (DrivingGPT, Doe-1, VaVAM) and related robot learning literature in the **General Response.** Following your suggestion, we have also revised the **Introduction** and **Related Work** to explicitly acknowledge these foundational contributions and better articulate our specific positioning.
>
> ### W2
>
> Efficiency concerns (train-inference gap and latency).
>
> ### R2
>
> We respectfully clarify that **bypassing image generation during inference does not create a train-inference gap**, as it strictly follows the standard autoregressive (next-token prediction) paradigm. During **training**, the model processes a full sequence (e.g., $I_{t-1}, A_{t-1}, I_t, A_t$) and computes loss on all tokens via causal masking. During **inference**, we simply "prefill" the context up to the current observation ($I_{t-1}, A_{t-1}, I_t$) and let the model predict the subsequent tokens, which correspond to $A_t$. Once the action tokens are generated, we terminate the process, thereby **avoiding** the high latency associated with generating future image tokens. To ensure causal correctness in the NAVSIM setting, we define the history action $A_{t-1}$ as a short **1s** trajectory and the current prediction target $A_t$ as the full **4s** planning trajectory, preventing information leakage while maintaining a consistent context format. Regarding the latency concern, as detailed in our main paper and Appendix B.1, this design ensures the total inference time remains within an acceptable range for real-time deployment (**74.3ms** with query-based action expert on an H200 GPU).
>
> ### W3
>
> Representation stability in the second training stage.
>
> ### R3
>
> This is an insightful question. Our empirical results indicate that **once the model is pretrained with the world modeling objective, retaining it during the fine-tuning stage yields negligible difference.** As shown in the table below, the performance gap is minimal (**85.6 PDMS** without WM loss vs. **85.4 PDMS** with it).  We attribute this robustness to the fact that robust visual representations are already established in the first stage. Furthermore, the second stage employs a **very short schedule** (4k steps, equivalent to roughly 2 epochs on NAVSIM), which prevents catastrophic forgetting of these pre-trained features even without explicit world modeling supervision.
>
> | Pretrain loss | Finetune loss | NC | DAC | PDMS |
> | --- | --- | --- | --- | --- |
> | WM + Action  | WM + Action | 98.3 | 93.8 | 85.4 |
> | WM + Action  |  Action | 98.3 | 93.9 | 85.6 |

---

### Official Review · Reviewer_kyet · 2025-11-09

**Soundness:** 3
**Presentation:** 3
**Contribution:** 3
**Rating:** 4
**Confidence:** 3

**Summary:**

This paper presents DriveVLA-W0, a Vision-Language-Action (VLA) model augmented with a world model to predict future visual frames. The goal is to provide denser self-supervised signals to compensate for the sparsity of action labels in end-to-end driving. Two variants are proposed: an autoregressive world model for discrete (VQ-tokenized) features and a diffusion-based world model for continuous visual features. The paper claims that the proposed method amplifies data-scaling behavior—i.e., performance continues to improve as data grows. Results on NAVSIM and a large-scale in-house dataset show consistent improvements in driving performance and sample efficiency.

**Strengths:**

* The empirical analysis of data scaling is thorough and carefully conducted across multiple model and data sizes.
* Both autoregressive and diffusion-based variants are implemented, illustrating generality across different architectures.
* The paper contributes valuable large-scale empirical findings to the community, especially around scaling trends and model efficiency.

**Weaknesses:**

* The core contribution is incremental. The main idea—adding a future-frame prediction objective as an auxiliary loss—is conceptually simple and has been explored in prior work on world models (e.g., WorldVLA, UniVLA).
* The proposed “world model” is used only as auxiliary supervision during training and not for downstream planning, simulation, or closed-loop control, limiting its conceptual impact.
* Although the paper introduces a “Mixture-of-Experts” Action Expert, the 8B VLA backbone is still required at inference. The MoE design primarily reduces the cost of the action decoding stage rather than removing the need for the large backbone itself, so the memory footprint and deployment complexity might remain dominated by the 8B model.
* Qualitative visualizations of predicted future frames with different actions are missing.

**Questions:**

* Can the learned world model predict future observations conditioned on different actions, and could it be leveraged for simulation or counterfactual reasoning?
* How sensitive is the performance to the weighting between the action-prediction and world-modeling losses?
* Could the authors include more qualitative visualizations or failure cases to better illustrate model behavior and limitations?

---

> ### Author Response · Authors · 2025-11-24
> **Response to Reviewer kyet (Part1)**
>
> We sincerely thank Reviewer kyet for the comprehensive review and the positive recognition of our thorough data scaling analysis and the generality of our approach across different architectures. We also deeply appreciate your constructive suggestions, which have helped us to improve the quality of our manuscript. Please find our detailed point-by-point responses below.
>
> ### W1
>
> Concerns regarding incremental contribution and relation to prior works.
>
> ### R1
>
> We sincerely thank the reviewer for raising this point regarding the scope of our contribution and its relation to prior works like WorldVLA and UniVLA. We have provided a detailed discussion clarifying our core contributions in the **General Response**. Following your constructive suggestion, we have also significantly revised the **Introduction** and **Related Work** sections in the updated paper to better articulate these distinctions and give full credit to these precedent studies.
>
> ### W2
>
> Other utility of the world model
>
> ### R2
>
> We thank the reviewer for this inspiring insight. Although primarily motivated by representation learning, we respectfully clarify that our learned model possesses inherent capabilities for simulation and planning, demonstrated in three aspects:
>
> - **Reactive Simulation (Figure 5):** The model functions as a reactive simulator, capable of simulating plausible future scenarios, such as the visual consequences induced by a "decelerate" action.
> - **Counterfactual Reasoning (Appendix C.4, Figure 14):** Conditioning on counterfactual actions (e.g., driving off-road) produces geometrically consistent future frames, validating the model's understanding of physical structure.
> - **Chain-of-Vision:** Our framework supports a "Chain-of-Vision" inference paradigm, analogous to Chain-of-Thought. Specifically, the model can hallucinate plausible future image sequences to guide trajectory generation. Currently, explicit image generation during inference is bypassed to ensure real-time efficiency. However, this inherent capability offers a promising path for closed-loop control as inference acceleration matures.
>
> ### W3
>
> Deployment feasibility and memory footprint of the 8B backbone.
>
> ### R3
>
> Thank you for this valuable question regarding deployment feasibility. We address this concern from two perspectives. **First**, for advanced hardware, the 8B model is well within acceptable limits. As detailed in the table below, total inference takes **~74.3ms** on an H200 GPU with a memory footprint of **~19GB**. While we acknowledge the compute gap between server-grade GPUs and edge devices, this is effectively bridged by industry-standard optimizations (e.g., INT8 quantization) and the adoption of next-generation automotive platforms like **NVIDIA Thor**. Even on current flagship chips like Orin-X (**up to 64GB memory**), the model fits comfortably within memory constraints. **Second**, for resource-constrained scenarios, our architecture allows for a seamless transition from 7B to smaller backbones like **Qwen2.5-VL (3B)**. As shown in the table, this variant significantly reduces system overhead while maintaining robust capability. Our experiments indicate only a marginal performance drop from **85.3 (7B) to 84.5 (3B)**, offering a highly efficient trade-off.
>
> | Base Model  | Component | Inference Memory | Inference Latency |
> | --- | --- | --- | --- |
> | Emu3 (8B) |  VLA Expert  (8B) | 17.73G | 66.3ms |
> |  | Action Expert  (500M) | 1.17G | 8.0ms |
> | Qwen2.5-VL (3B) |  VLA Expert (3B) | 6.64G | 46.6ms |
> |  | Action Expert (480M) | 1.02G | 6.8ms |

---

> ### Author Response · Authors · 2025-11-24
> **Response to Reviewer kyet (Part2)**
>
> ### W4 & Q1
>
> Capabilities for action-conditioned generation and simulation.
>
> ### Response to W4&Q1
>
> We address these points together as they both concern the model's ability to perform action-conditioned generation. We have added comprehensive visualizations in **Figure 5 and Appendix C.4 (Figure 14)** to explicitly validate this capability:
>
> - **Reactive Simulation (Figure 5):** Demonstrates the model functioning as a reactive simulator, capable of simulating plausible visual consequences induced by a "decelerate" action.
> - **Counterfactual Reasoning (Figure 14):** Shows that a counterfactual "turn right" command.
>
> ### Q2
>
> Sensitivity to loss weights.
>
> ### R2
>
> To assess sensitivity, we conducted ablation studies on the world modeling loss weight ($\alpha$). Due to the limited duration of the rebuttal phase, we performed this specific ablation by training from scratch on the NAVSIM dataset, rather than repeating the full NuPlan pre-training pipeline. As shown in the table below, **the performance is remarkably robust to weight variations once the world modeling objective is introduced.** While the baseline ($\alpha=0$) achieves only **68.7 PDMS**, adding even a small weight ($\alpha=0.1$) yields a substantial leap to **82.5 PDMS**, demonstrating the critical value of the auxiliary supervision. Performance peaks at **83.8 PDMS** with $\alpha=0.5$ and remains stable at $\alpha=1.0$ (**83.0 PDMS**), indicating that the system is not hypersensitive to hyperparameter tuning within a reasonable range.
>
> | WM loss weight | Action loss weight | NC | DAC | PDMS |
> | --- | --- | --- | --- | --- |
> | 0 | 1 | 93.3 | 80.9 | 68.7 |
> | 0.1 | 1 | 97.6 | 91.6 | 82.5 |
> | 0.5 | 1 | 98.0 | 92.5 | 83.8 |
> | 1 | 1 | 97.6 | 91.8 | 83.0 |
>
> ### Q3
>
> Failure Case Analysis.
>
> ### R3
>
> We appreciate this suggestion to provide a balanced view of our model's performance. We have included a detailed **Failure Case Analysis in Appendix C.2 (Figures 11 & 12)**, categorizing limitations into two primary types:
>
> - **Instruction Ambiguity (Figure 11):** We illustrate how coarse-grained commands (e.g., "go straight") cause ambiguity at complex topologies like Y-junctions, leading to planning indecision.
> - **Dynamic Prediction Errors (Figure 12):** We analyze a complex intersection scenario where the model fails to anticipate an oncoming vehicle, resulting in an unsafe left turn.

---

### Author Response · Authors · 2025-11-24
**General Response**

We sincerely thank all reviewers for their comprehensive reviews and constructive feedback, which have significantly helped us improve the quality of our manuscript. In addition to our detailed point-by-point responses below, we would like to address a common concern regarding the novelty of our approach compared to prior art.

**Concern About Novelty**:

Several reviewers noted that using future prediction as an auxiliary loss is not a new concept (citing WorldVLA, UniVLA, DrivingGPT, etc.) and questioned the incrementality of the contribution.

**Response:**

We appreciate the reviewers pointing out these related works. While we agree that the *concept* of world modeling exists in the literature, DriveVLA-W0 introduces two critical innovations that distinguish it from prior arts and address fundamental limitations.

**1. Diffusion World Model for Continuous VLMs**

**Prior VLA works that incorporate world modeling (e.g., WorldVLA, UniVLA, DrivingGPT) rely exclusively on Autoregressive (AR) paradigms.** This approach necessitates discretizing images into tokens. Such a design is incompatible with modern high-performance VLMs (e.g., Qwen2.5-VL), which inherently operate on continuous visual features without a discrete vocabulary. To bridge this gap, we propose a **Diffusion World Model** specifically designed to provide dense, self-supervised signals for ViT-based architectures. This contribution significantly extends the scope of world modeling beyond VQ-based frameworks, allowing mainstream continuous VLMs to benefit from dense supervision.

**2.  Novel Discoveries at Massive Data Scale**

**Beyond architecture design, we systematically uncover critical data scaling behaviors often obscured in current benchmarks.** A common pitfall in deep learning is that techniques effective in low-data regimes often diminish as data scales. Our work leverages massive-scale data to uncover three critical phenomena unobservable on smaller benchmarks. **First**, we find that world modeling does not merely provide a static boost but increases the **slope of the scaling law**, accelerating gains where action-only supervision saturates. **Second**, we observe a **"performance reversal"** in action decoding as the data scale expands, where simpler autoregressive models surpass complex flow-matching baselines in high-data regimes. **Third**, we demonstrate that true generalization is not achieved by fitting action distributions (which leads to overfitting) but requires learning **universal visual representations** via world modeling.

We have summarized these differences in the table below:

|  | World Model | Dataset Scale（hrs) | Domain | Dataset |
| --- | --- | --- | --- | --- |
| WorldVLA [1] | AR | ~27 | Robot | LIBERO |
| UniVLA [2] | AR | ~2,821 | Robot | RT-1,DROID etc |
|  LAW [3] | Latent | ~5.5 | Driving | nuscenes |
| DrivingGPT [4] | AR | ~128 | Driving | nuPlan, navsim |
| Doe-1 [5] | AR | ~5.5 | Driving | nuScenes |
| VaVAM [6] | AR | ~1800 | Driving | OpenDV, nuPlan, nuScenes |
| DriveVLA-W0 (ours) | AR & Diffusion | ~8400 | Driving | Inhouse Dataset , nuplan |

[1]: WorldVLA: Towards Autoregressive Action World Model (Arxiv 2025)

[2]: Unified Vision-Language-Action Model (Arxiv 2025)

[3]: Enhancing End-to-end Autonomous Driving with Latent World Model (ICLR 2025)

[4]: DrivingGPT: Unifying Driving World Modeling and Planning with Multi-modal Autoregressive Transformers (ICCV 2025)

[5]: Doe-1: Closed-Loop Autonomous Driving with Large World Model   (Arxiv 2025)

[6]: VaViM and VaVAM: Autonomous Driving through Video Generative Modeling (Arxiv 2025)

---

### Author Response · Authors · 2025-11-24
**Revision of the PDF file**

We have uploaded a revised version of the paper. Beyond the point-by-point responses, we would like to highlight the major improvements made during the rebuttal phase:

**1. Textual Revisions (Intro & Related Work):**
We revised the Introduction to tone down claims. We also significantly expanded the Related Work section to clarify the novelty of our work compared to recent world modeling literature.

**2. Comprehensive Visualizations:**
We added several key figures to qualitatively assess the model:

- **Figure 5 (Action-Conditioned Simulation):** We demonstrate the model's capability for action-conditioned image generation, highlighting its potential for reactive simulation.
- **Figure 11 & 12 (Failure Cases):** We honestly analyze current limitations regarding instruction ambiguity and dynamic object prediction.
- **Figure 14 (Counterfactual Reasoning):** We show how the model hallucinates reasonable futures (e.g., driving off-road) given counterfactual actions.

**3.Enhanced Closed-loop Evaluation (NAVSIMv2):**
We introduced NAVSIMv2 into our evaluation suite. This inclusion supports reactive agents, providing a closer approximation to real-world closed-loop driving.

---

### Author Response · Authors · 2025-12-03
**Rebuttal Summary for Area Chair**

Dear Area Chair,

We sincerely thank you and the reviewers for the time and effort dedicated to reviewing our work. We have received 4 reviews with current ratings of **8, 6, 4, 4**. We are particularly encouraged that **Reviewer uuy7 (Score: 8)** actively engaged during the discussion period, confirming that our response and additional experiments effectively resolved their concerns and maintaining their strong support.

Below is a summary of our rebuttal updates and responses to key concerns:

**1. General Response: Novelty & Contribution**
A shared concern among reviewers was the novelty of our framework compared to prior arts (e.g., WorldVLA, UniVLA). We clarified that **DriveVLA-W0** distinguishes itself through two core contributions:

- **Methodological Innovation:** We propose a **Diffusion World Model** specifically for continuous ViT-based VLMs, addressing the limitation of prior AR-only approaches that are incompatible with modern continuous architectures (e.g., Qwen2.5-VL).
- **Scaling Insights:** Our large-scale investigation reveals that world modeling acts as a "catalyst" to increase the slope of the scaling law, accelerating gains where action-only supervision saturates. Furthermore, we prove that true generalization relies on learning universal visual representations via world modeling rather than merely fitting action distributions.

 We have significantly revised the **Introduction** and **Related Work** to explicitly articulate these distinctions.

**2. Summary of Specific Reviewer Responses**

- **Reviewer uuy7 (Score: 8):**
    - **Strengths:** Praised the paper for being well-motivated with insightful analysis and strong performance.
    - **Concern 1 (Training Cost):** We provided a detailed cost analysis showing that the training overhead of world modeling is minimal.
    - **Concern 2 (Sensor Generalization):** We added new experiments with multi-view inputs, demonstrating that the benefits of world modeling persist and amplify performance even with richer sensor suites.
- **Reviewer dcaL (Score: 6):**
    - **Strengths:** Recognized our rigorous experimental design and clear motivation.
    - **Concern 1 (Performance Anomaly):** We identified that a specific baseline underperformance was due to a scheduler configuration. We provided updated experimental results with the corrected schedule, confirming that our method consistently outperforms the baseline.
    - **Concern 2 (Baselines):** We added **ReCogDrive** [1] (a recent VLA method) as a strong baseline on our in-house dataset to ensure a more complete comparison.
    - **Concern 3 (Closed-loop):** We expanded evaluations to include **NAVSIM v2** [2] (reactive agents) and our internal closed-loop simulator, achieving an **MPI** (Miles Per Intervention) > **3km**.
- **Reviewer 83sR (Score: 4):**
    - **Strengths:** Acknowledged the experimental thoroughness and clear writing.
    - **Concern 1 (Inference Efficiency):** The reviewer may have assumed that image generation is required during inference. We clarified that image generation is bypassed during deployment, ensuring high speed (74ms) with no train-inference gap.
    - **Concern 2 (Two-stage Training):** We explained and validated via ablation studies that world modeling loss is not strictly necessary during the short fine-tuning stage (Stage 2), as the robust representations acquired in Stage 1 are preserved.
- **Reviewer kyet (Score: 4):**
    - **Strengths:** Appreciated the scaling law insights and architectural generality.
    - **Concern 1 (Generative Capability):** The reviewer expressed concern that our world model was limited to auxiliary supervision. We demonstrated its capability to explicitly **generate action-conditioned images** by adding 6 new visualizations into the revised paper, including action-conditioned simulation, counterfactual reasoning, and failure case analysis.
    - **Concern 2 (Efficiency):** We addressed concerns about deployment feasibility by providing a detailed breakdown of latency and memory footprint.

We believe the additional experiments (Multi-view, Closed-loop, New Baselines) and visualizations significantly strengthen the paper. We hope this summary assists in your final assessment.

Best regards,

Authors of Paper #349

[1]: ReCogDrive: A Reinforced Cognitive Framework for End-to-End Autonomous Driving

[2]: Pseudo-Simulation for Autonomous Driving

---

### Meta-Review · Area_Chair_tH7J · 2025-12-29

**Summary:**

The paper enhances VLA models for autonomous driving by introducing action-conditioned future-image prediction as an auxiliary prediction task to address the issue of sparse action labels. The reviewers generally agree that the paper, especially its empirical analyses, is well executed and valuable.

**Reviewer Concerns:**

A major concern is in the novelty of the main idea, that using next-state prediction as an auxiliary task to enhance representation learning is a well-known idea. The authors have acknowledged the related prior works in the revision and clarified the paper's contributions in light of them.

**Reviewer Scores:**

The scores are 8/6/4/4. Other than the 8, other reviewers have not had the chance to respond to the rebuttal.

---

### Decision · Program_Chairs · 2026-01-26

Accept (Poster)